# Neural Tangent Kernel Maximum Mean Discrepancy

**Xiuyuan Cheng**
Department of Mathematics
Duke University
xiuyuan.cheng@duke.edu

**Yao Xie**
H. Milton Stewart School of Industrial
and Systems Engineering
Georgia Institute of Technology
yao.xie@isye.gatech.edu

## Abstract

We present a novel neural network Maximum Mean Discrepancy (MMD) statistic by identifying a new connection between neural tangent kernel (NTK) and MMD. This connection enables us to develop a computationally efficient and memory-efficient approach to compute the MMD statistic and perform NTK based two-sample tests towards addressing the long-standing challenge of memory and computational complexity of the MMD statistic, which is essential for online implementation to assimilating new samples. Theoretically, such a connection allows us to understand the NTK test statistic properties, such as the Type-I error and testing power for performing the two-sample test, by adapting existing theories for kernel MMD. Numerical experiments on synthetic and real-world datasets validate the theory and demonstrate the effectiveness of the proposed NTK-MMD statistic.

## 1   Introduction

Maximum Mean Discrepancy (MMD) statistic is a popular method in machine learning and statistics. In particular, kernel MMD [2, 23] has been applied to evaluating and training neural network generative models [34, 44, 31, 3, 30]. Though a widely used non-parametric test [23], kernel MMD encounters several challenges in practice. The roadblocks for large-scale implementation of kernel MMD involve heavy memory requirement (due to the computation and storage of the Gram matrix, which grows quadratically with the data size) and the choice of a good kernel function for high dimensional data. While Gaussian RBF kernel was shown to provide a metric between pairs of probability distributions with infinite data samples, applying isotropic Gaussian kernel to data in applications, such as image data and discrete events data, may invoke issues in terms of kernel expressiveness [25, 29, 36] and sampling complexity [38].

A potential path forward in developing more computationally and memory-efficient testing statistics is to leverage deep neural networks' representation and optimization advantage. For example, the idea of training a classification neural network for testing problems has been revisited recently in [37, 10], and the connection between classification and two-sample testing dates back to earlier works [19, 43, 40]. However, in applying deep models to testing problems, the test consistency analysis is usually incomplete due to the lack of optimization guarantee of the trained network. For one thing, assuming perfect training of a deep network to achieve global minimizer is too strong an assumption to fulfill in practice.

A recent focus of neural network optimization research is the so-called lazy training regime of over-parametrized neural networks [13], where the neural network training dynamics exhibit certain linearized property and provable learning guarantee can be obtained [33, 17, 15, 1]. In this regime, the training time is sufficiently short, and networks are sufficiently parametrized such that network parameters stay close to the randomized initial values over the training process. In particular, the Neural Tangent Kernel (NTK) theory, as firstly described by [28], shows that the network optimization can be well approximated by the Reproducing Kernel Hilbert Space (RKHS) formulation. The NTK

35th Conference on Neural Information Processing Systems (NeurIPS 2021).

theory has been developed for general neural network architectures, including deep fully connected networks [48], convolutional networks [6, 35], graph neural networks [16], and residual networks [45, 27, 7]. The RKHS approach by NTK has been shown theoretically and empirically to characterize the wide neural network training dynamic in the early stage.

The current work stems from a simple observation that short-time training of a network is approximately equivalent to computing the *witness function* of a kernel MMD with NTK at time zero, when the training objective equals the difference between sample averages of the network function on two samples. The proposed test statistic, called NTK-MMD, approximates the classical kernel MMD with NTK, and the error at training time $t$ can be bounded to be $O(t)$ under the linearization of the NTK theory. The theoretical benefit of translating the network-based statistic into a kernel MMD is that the testing power of the latter can be analyzed based on previous works. Algorithm-wise, the network-based test statistic can be computed on the fly: thanks to the form of linear accumulation of the training objective, the training allows small-batch, e.g., batch size 1 and 1 epoch of training (1 pass of the samples), under the NTK approximation. (The "NTK approximation" in this paper refers to approximating the time-$t$ NTK by the time-zero kernel, both at finite width.) To calibrate the testing threshold needed to prevent false alarm, we introduce an asymmetric MMD using training-testing split and theoretically prove the testing power, where the threshold is estimated from bootstrapping on the test split only and thus avoids retraining network.

Our main contributions include the following: (i) We introduce a neural network-based test statistic called NTK-MMD, which can be computed by a short-time training of a neural network, particularly online learning using one-pass of the training samples and batch size one. The NTK approximation error of the MMD statistic is shown to be $O(t)$, that is, linear in training time, and the result extends to Stochastic Gradient Descent training; (ii) We characterize the statistical properties of the NTK-MMD, including the Type-I error and the testing power, which establish the conditions under which the test is powerful; we further introduce a data split scheme such that the test threshold can be estimated without network retraining with provable testing power guarantee; (iii) The efficiency of the proposed NTK-MMD test is demonstrated on simulated and real-world datasets.

At the same time, we are aware of the limitations of NTK in explaining deep network optimization, expressiveness power, and so on. We discuss limitations and extensions in the last section. In particular, this paper focuses on demonstrating the power of NTK-MMD statistics for the two-sample test, while the proposed computationally and memory-efficient NTK-MMD statistics can also be used for other applications of MMD statistics [24] and hypothesis tests.

## 2 Method

### 2.1 Preliminary: Kernel MMD

We start by reviewing a few preliminaries. Consider data in $\mathcal{X} \subset \mathbb{R}^d$, sampled from two unknown distributions with densities $p$ and $q$. Given two data sets

$$X = \{x_i \sim p, \text{ i.i.d., } i = 1, \cdots, n_X\}, \quad Y = \{y_j \sim q, \text{ i.i.d., } j = 1, \cdots, n_Y\}, \quad (1)$$

we would like to test whether or not they follow the same distribution. This is equivalent to perform the following hypothesis test $H_0 : p = q$ versus $H_1 : p \neq q$. The classical kernel MMD considers test functions in the RKHS of positive semi-definite kernel $K(x, y)$, which can be, for instance, the Gaussian RBF kernel. The (squared and biased) empirical kernel MMD statistic is given by [23]:

$$\text{MMD}_K^2 = \int_{\mathcal{X}} \int_{\mathcal{X}} K(x,y)(\hat{p}-\hat{q})(x)(\hat{p}-\hat{q})(y)dxdy, \quad \hat{p} := \frac{1}{n_X}\sum_{i=1}^{n_X}\delta_{x_i}, \quad \hat{q} := \frac{1}{n_Y}\sum_{i=1}^{n_Y}\delta_{y_i}. \quad (2)$$

The null hypothesis is rejected if $\text{MMD}_K^2 > t_{\text{thres}}$, where $t_{\text{thres}}$ is the user-specified test threshold (usually, chosen to control the false alarm up to certain level). The (empirical) *witness function* of the MMD statistic, $\hat{w}(x) = \int_{\mathcal{X}} K(x,y)(\hat{p} - \hat{q})(y)dy$, indicates where the two densities differ. The Type-I error of the test is defined as $\mathbb{P}[\text{MMD}_K^2 > t_{\text{thres}}]$ under $H_0$, and the Type-II error as $\mathbb{P}[\text{MMD}_K^2 \leq t_{\text{thres}}]$ under $H_1$; the power is defined as one minus the Type-II error. For an alternative distribution $q$ of $p$, the test errors depend on $q$, the sample sizes $n_X$ and $n_Y$, as well as the kernel function $K(x, y)$. Theoretically, the test power of kernel MMD has been analyzed in [23], investigated for high dimensional Gaussian data in [38], and for manifold data in [12].

### 2.2 NTK-MMD statistic

As the proposed NTK-MMD framework can be used on different network architectures, we write the neural network mapping abstractly as $f(x; \theta)$, which maps from input $x \in \mathcal{X}$ to $\mathbb{R}$, and $\theta$ is the

network parameters. Use $X \cup Y$ as the training dataset, and let $\hat{p}$ and $\hat{q}$ be as in (2), we choose a particular training objective function as

$$\hat{L}(\theta) = -\int_{\mathcal{X}} f(x;\theta)(\hat{p}-\hat{q})(x)dx = -\frac{1}{n_X}\sum_{i=1}^{n_X} f(x_i;\theta) + \frac{1}{n_Y}\sum_{i=1}^{n_Y} f(y_i;\theta) \qquad (3)$$

The choice of this objective function is critical in establishing the connection between NTK and MMD. Optimizing this objective will lead to divergence of the network function if we train for a long time. However, if the training is only for a short time, the network function remains finite. Here we mainly focus on short-time training of the network, and particularly the online training setting where the number of epochs is 1, that is, only 1-pass of the training samples is used. We will also show that the method allows using minimal batch size for online learning without affecting the NTK approximation of the MMD statistic, c.f. Remark 2.2.

Following the convention in NTK literature, below we formulate in terms of continuous-time Gradient Descent (GD) dynamic of the network training. The extension to discrete-time Stochastic Gradient Descent (SGD) holds with a small learning rate (Remark 2.2). The network parameter $\theta(t)$ evolves according to $\dot{\theta}(t) = -\partial\hat{L}/\partial\theta$, and we define $u(x,t) := f(x,\theta(t))$, which is the network mapping function at time $t$. Suppose the network is trained for a short time $t > 0$, we define

$$\hat{g}(x) := \frac{1}{t}(u(x,t) - u(x,0)), \qquad (4)$$

and the test statistic, which depends on time $t$, is

$$\hat{T}_{\mathrm{net}}(t) := \int_{\mathcal{X}} \hat{g}(x)(\hat{p}-\hat{q})(x)dx = \frac{1}{t}\left(\hat{L}(\theta(t)) - \hat{L}(\theta(0))\right). \qquad (5)$$

The function $\hat{g}$ is the difference of the network mapping after a short-time training from the initial one, and we call it the *witness function* of network NTK-MMD statistic. As revealed by (5), (without calibrating the test threshold) the test statistics $\hat{T}_{\mathrm{net}}$ is nothing but the decrease in the training objective, and comes as a by-product of network training at no additional computational cost. We show in next subsection that at small $t$, the statistic $\hat{T}_{\mathrm{net}}(t)$ provably approximates the classical MMD statistic with the NTK, i.e. $\hat{T}_{\mathrm{NTK}} = \mathrm{MMD}_K^2(X,Y)$ where $K$ is the NTK at time $t = 0$ as in (8). Algorithmically, we will perform two-sample test using $\hat{T}_{\mathrm{net}}(t)$ by comparing with a threshold $t_{\mathrm{thres}}$.

## 2.3 NTK approximation of MMD statistic

In the continuous-time training dynamic of the network, we consider the NTK [28] kernel function defined for $t > 0$ as

$$\hat{K}_t(x,x') := \langle \nabla_\theta f(x;\theta(t)), \nabla_\theta f(x';\theta(t))\rangle. \qquad (6)$$

The following lemma follows directly by construction, and the proof is in Appendix A.1.

**Lemma 2.1.** *The network function $u(x,t)$ satisfies that for $t > 0$,*

$$u(x,t) - u(x,0) = \int_0^t \int_{\mathcal{X}} \hat{K}_s(x,x')(\hat{p}-\hat{q})(x')dx'ds. \qquad (7)$$

It has been shown (in [6, 5, 13], among others) that for the short-time training (lazy training regime), the kernel (6) can be well-approximated by the kernel at time $t = 0$, namely

$$K_0(x,x') := \langle \nabla_\theta f(x;\theta(0)), \nabla_\theta f(x';\theta(0))\rangle, \qquad (8)$$

which is only determined by the network weight initialization $\theta(0)$. Assuming $K_0 \approx \hat{K}_t$ in Lemma 2.1, the proposed test statistic $\hat{T}_{\mathrm{net}}(t)$ as in (5) can be viewed as

$$\hat{T}_{\mathrm{net}}(t) \approx \hat{T}_{\mathrm{NTK}} := \int_{\mathcal{X}}\int_{\mathcal{X}} K_0(x,x')(\hat{p}-\hat{q})(x)(\hat{p}-\hat{q})(x')dxdx', \qquad (9)$$

which is the kernel MMD statistic with NTK. (See Remark A.1 for a discussion on biased/unbiased MMD estimator.) In below, we show in Proposition 2.1 that the approximation $\hat{T}_{\mathrm{net}} \approx \hat{T}_{\mathrm{NTK}}$ has $O(t)$ error, and we experimentally verify the similarity of the two statistics in Subsection 4.2. Throughout the paper, we compute $\hat{T}_{\mathrm{net}}$ by neural network training, and we call $\hat{T}_{\mathrm{NTK}}$ the *exact* NTK-MMD

which is for theoretical analysis. The theoretical benefit of translating $\hat{T}_{\text{net}}$ into $\hat{T}_{\text{NTK}}$ lies in that testing power analysis of $\hat{T}_{\text{NTK}}$ follows existing methods which is detailed in Section 3.

Suppose neural network parameter $\theta$ is in $\mathbb{R}^M$ and $\theta \in \Theta$, where $\Theta$ is a domain in $\mathbb{R}^M$ which contains the Euclidean ball $B(\theta(0), r_0)$, where we assume $r_0$ is an $O(1)$ constant. For vector valued function $g : (\mathcal{X}, \Theta) \to \mathbb{R}^d$ and $U \subset \Theta$, we denote the infinity norm as $\|g\|_{\mathcal{X}, U} := \sup_{x \in \mathcal{X}, \theta \in U} \|g(x, \theta)\|$. When $g$ maps to a matrix, the notation denotes (the infinity norm over $(\mathcal{X} \times U)$ of) the operator norm. The test statistic approximation error in Proposition 2.1 directly follows the following lemma concerning the uniform approximation of the kernels. All proofs in Appendix A.1.

**Lemma 2.2** (NTK kernel approximation). *Suppose $f$ is $C^2$ on $(\mathcal{X}, \Theta)$ and $\|\nabla_\theta f\|_{\mathcal{X}, \Theta} \leq L_f$ for some positive constant $L_f$. Then for any $0 < r < r_0$, when $0 < t < t_{f,r} := r/(2L_f)$,*
*(1) $\theta(t)$ stays inside the Euclidean ball $B_r := B(\theta(0), r)$.*
*(2) Define $C_{f,r} := 4\|D_\theta^2 f\|_{\mathcal{X}, B_r} \|\nabla_\theta f\|_{\mathcal{X}, B_r}^2$, we have that*

$$\sup_{x, x' \in \mathcal{X}} |\hat{K}_t(x, x') - K_0(x, x')| \leq C_{f,r} t. \tag{10}$$

*Remark* 2.1 (Boundedness of $\|\nabla_\theta f\|_{\mathcal{X}, \Theta}$). When $p$ are unbounded density (gaussian), and activation function $f$ is relu or softplus, the uniform boundednesss of $\|\nabla_\theta f\|_{\mathcal{X}, \Theta}$ may fail. However, for sub-exponential densities, apply standard truncation argument, and when we restrict to compactly supported distributions. In practice, we standardize the data to be on a compact domain in $\mathbb{R}^d$.

**Proposition 2.1** (Test statistic approximation). *The condition on $f(x, \theta)$ is the same as in Lemma 2.2, and for $0 < r < r_0$, the constants $t_{f,r}$ and $C_{f,r}$ are as therein. Then, when $0 < t < t_{f,r}$, we have that*

$$|\hat{T}_{\text{net}}(t) - \hat{T}_{\text{NTK}}| \leq 2C_{f,r} t.$$

*Remark* 2.2 (SGD and online training). The above error bound analysis based on Taylor expansion can extend to discrete-time GD dynamic by showing that the time discretization introduces higher-order error when $t$ is small. In the SGD setting, e.g., the online learning of 1 epoch, batch size one, and learning rate $\alpha$, the network parameters are updated after scanning each training sample on the fly. Let $\theta_k$ be the network after scanning $k$ many samples, we show in Appendix A.2 that the difference $\|\theta_k - \theta_0\|$ can be bounded by $O(\alpha k/n)$, and the trained network witness function after 1 epoch approximates the witness function with the zero-time NTK kernel up to an $O(\alpha)$ error. The learning rate $\alpha$ has the role of training time $t$. The fact that batch size will not affect the NTK approximation of the network training is a result of that the loss (3) is a linear accumulation over samples, which may not hold for other loss types. The compatibility with online learning and training with very small batch size of NTK-MMD statistic makes it convenient for deep network training, especially under memory constraints.

### 2.4 Computational and memory efficiency

The update of network parameters in NTK-MMD training can be viewed as an implicit computation of the inner-product between high dimensional kernel feature maps $\langle \nabla_\theta f(x; \theta(t)), \nabla_\theta f(x'; \theta(t)) \rangle$ (by chain rule, c.f. (24) (25) in Appendix A.1). The network witness function $\hat{g}$ defined in (4) is parametrized and stored in trained network parameters. This allows the (approximate) evaluation of kernel on a test sample $x'$ without computing the gradient $\nabla_\theta f(x'; \theta)$ explicitly. It also means that the NTK network witness function can be evaluated on any new $x'$ without revisiting the training set. In contrast, traditional kernel MMD computes kernel witness function (defined as $\int K(x, y)(\hat{p} - \hat{q})(y)dy$ [23]) on a new point $x'$ by pairwise computation between $x'$ and samples in datasets $X$ and $Y$.

NTK-MMD can be computed via batch-size-one training over one-pass of the training set (c.f. Remark 2.2 and experimentally verified in Table A.2). The gradient field evaluation (back propagation) is only conducted on the training set but not the testing test, and the bootstrap calibration of the test threshold can be computed from test set only (c.f. Section 3.2). Thus, by using small learning rate (allowed by floating point precision, c.f. Remark C.1), one can incorporate large number of training samples via more training iterations without worsening the approximation error to exact NTK-MMD, which will improve testing power. This "separation" of training and testing, in memory and computation, of NTK-MMD allows scalable online learning as well as efficient deployment of the network function on potentially large test sets.

## 3 Theoretical properties of NTK-MMD

In this section, we prove the testing power (at a controlled test level) of the NTK-MMD statistic $\hat{T}_{\text{NTK}}$ as in (9) with large enough finite samples. We also introduce an asymmetric version of the

MMD statistic using training-testing dataset splitting, which enables the bootstrap estimation of the threshold of the test $t_{\text{thres}}$ without retraining of the neural network.

## 3.1 NTK-MMD without data splitting

We write the NTK kernel $K_0(x, x')$ as $K(x, x')$ omitting the subscript, and assume that $K(x, x')$ is uniformly bounded, that is, $\sup_{x \in \mathcal{X}} K(x, x) \leq B < \infty$ for some positive constant $B$. Without loss of generality, we assume that $B = 1$ (because a global constant normalization of the kernel introduces a global constant multiplied to the test statistic, and does not change the testing). By that the kernel is PSD, we thus have that

$$\sup_{x', x \in \mathcal{X}} |K(x, x')| \leq 1. \tag{11}$$

We omit the $_{\text{NTK}}$ subscript and denote the exact NTK-MMD statistic in (9) as $\hat{T}$. The corresponding population statistic is the squared MMD of kernel $K(x, y)$

$$\delta_K := \text{MMD}_K^2(p, q) = \int_{\mathcal{X}} \int_{\mathcal{X}} K(x, y)(p - q)(x)(p - q)(y)dxdy. \tag{12}$$

By the uniform boundedness (11), the kernel $K(x, x')$ is in $L^2(\mathcal{X} \times \mathcal{X}, (p+q)(x)(p+q)(x')dxdx')$. We define the squared integrals of the kernel

$$\nu_{pp} := \mathbb{E}_{x \sim p, y \sim p} K(x, y)^2, \quad \nu_{pq} := \mathbb{E}_{x \sim p, y \sim q} K(x, y)^2, \quad \nu_{qq} := \mathbb{E}_{x \sim q, y \sim q} K(x, y)^2. \tag{13}$$

In addition, we assume that as $n := n_X + n_Y$ increases, $n_X/n$ stay bounded and approaches $\rho_X \in (0, 1)$. Equivalently, there is some constant $0 < c < 1$ such that for large enough $n$,

$$cn + 1 \leq n_X, n_Y \leq n, \quad i = 1, 2. \tag{14}$$

Without loss of generality, we assume that (14) always holds for the $n$ considered.

**Theorem 3.1** (Test power of $\hat{T}_{\text{NTK}}$)**.** *Suppose (11) and (14) hold, and*

*(i) Under $H_1$, $p \neq q$, the squared population kernel MMD $\delta_K$ as in (12) is strictly positive,*

*(ii) The three integrals as in (13), $\nu_{pp}, \nu_{pq}, \nu_{qq}$, are all bounded by a constant $\nu \leq 1$.*

*Define $\lambda_1 := \sqrt{8 \log(4/\alpha_{\text{level}})}$, and let the threshold for the test be $t_{\text{thres}} = 4/(cn) + 4\lambda_1 \sqrt{\nu/cn}$. Then, if for some $\lambda_2 > 0$, $n$ is large enough such that*

$$n > \frac{1}{c} \max \left\{ \frac{1}{9\nu} \max\{\lambda_1, \lambda_2\}^2, \frac{8}{\delta_K}, \frac{\nu}{\delta_K^2} (8(\lambda_1 + \lambda_2))^2 \right\}, \tag{15}$$

*then under $H_0$, $\mathbb{P}[\hat{T} > t_{\text{thres}}] \leq \alpha_{\text{level}}$; and under $H_1$, $\mathbb{P}[\hat{T} \leq t_{\text{thres}}] \leq 3e^{-\lambda_2^2/8}$.*

The proof uses the U-statistic concentration analysis, and is left to Appendix B. As revealed by the proof, the diagonal entries in the kernel matrix contribute to the $O(1/n)$ term, and thus switching from the biased estimator of MMD (9) to the unbiased estimator gives similar theoretical results.

*Remark* 3.1 (Choice of $t_{\text{thres}}$). The choice of $t_{\text{thres}}$ in the above theorem is a theoretical one and may not be optimal, due to the use of concentration inequality and the relaxation of the bounds by using constants $\nu$ and $c$. By definition, the optimal value of $t_{\text{thres}}$ is the $(1-\alpha_{\text{level}})$-quantile of the distribution of $\hat{T}$ under $H_0$. The asymptotic choice may be obtained analytically according to the limiting distribution of the MMD statistic, c.f. Remark B.1. The threshold $t_{\text{thres}}$ is also computed by a bootstrap strategy in practice [4] (called "full-bootstrap" in next subsection). The bootstrap approach permutes the labels in data sets $X$ and $Y$, and since in $\hat{T}_{\text{net}}$ the witness function $\hat{g}(x)$ is computed by neural network training, this will incur retraining of the network. A solution to avoid retraining by adopting a test set for bootstrap estimation of $t_{\text{thres}}$ is introduced in next subsection.

## 3.2 Threshold calibration by data splitting

As shown in Theorem 3.1 and Remark 3.1, in the theoretical characterization of test power (at a required test level) the test threshold plays a critical role. In practice, we need a more precise threshold to exactly control the false alarm under the null hypothesis. In this section, we discuss how to set the threshold in two settings: fixed-sample and pilot-data. Nevertheless, we would like to mention that there exist applications where the threshold is not needed, and the symmetric MMD (without training/test split) can be used as a measurement of distribution divergence.

**Fixed-sample setting.** We first consider the setting where we have a fixed number of samples from $p$ and $q$. To obtain a precise threshold to control the false alarm, we need to split data into two non-overlapping parts: one part for training neural networks (compute the witness function) and one part data for bootstrapping and calibrating the threshold. We want to highlight that here we develop a scheme for threshold calibration such that no retraining of the witness function is necessary.

We randomly split the datasets $X$ and $Y$ into training and testing sets, $X = X_{(1)} \cup X_{(2)}$ and $Y = Y_{(1)} \cup Y_{(2)}$, and compute an asymmetric version of kernel MMD (the subscript $_a$ is for "asymmetric")

$$\hat{T}_a := \int_{\mathcal{X}} \int_{\mathcal{X}} K(x, x')(\hat{p}_{(1)} - \hat{q}_{(1)})(x')(\hat{p}_{(2)} - \hat{q}_{(2)})(x) dx dx', \tag{16}$$

where $\hat{p}_{(i)}$ and $\hat{q}_{(i)}$ are the empirical measures of datasets $X_{(i)}$ and $Y_{(i)}$ respectively, $i = 1, 2$. Define $n_{X,(i)} = |X_{(i)}|$ and $n_{Y,(i)} = |Y_{(i)}|$, $i = 1, 2$. Similarly as in Section 2, the MMD statistic (16) with $K(x, y) = K_0(x, y)$, the zero-time NTK, can be approximated by

$$\hat{T}_{a,\text{net}}(t) = \int_{\mathcal{X}} \hat{g}_{(1)}(x)(\hat{p}_{(2)} - \hat{q}_{(2)})(x) dx, \quad \hat{g}_{(1)}(x) = \frac{1}{t}(\hat{u}(x, t) - \hat{u}(x, 0)), \tag{17}$$

for a small time $t$, where $\hat{u}(x, t)$ is the network function trained by minimizing $\hat{L}(\theta) := -\int_{\mathcal{X}} f(x; \theta)(\hat{p}_{(1)} - \hat{q}_{(1)})(x) dx$ on the training set $\mathcal{D}_{tr} = \{X^{(1)}, Y^{(1)}\}$ with binary labels $\{1, 2\}$. Same as in Lemma 2.2 Proposition 2.1, the difference $|\hat{T}_a - \hat{T}_{a,\text{net}}(t)|$ can be bounded to be $O(t)$. We theoretically analyze the testing power of $\hat{T}_a$ where $K(x, y) = K_0(x, y)$ in below.

The benefit of splitting the test set lies in that once the witness function $\hat{g}_{(1)}(x)$ is trained from $\mathcal{D}_{tr}$, one can do a *test-only bootstrap* which is to compute

$$\hat{T}_{a,\text{null}} = \int_{\mathcal{X}} \hat{g}_{(1)}(x)(\hat{p}'_{(2)} - \hat{q}'_{(2)})(x) dx, \tag{18}$$

where $\hat{p}'_{(2)}$ and $\hat{q}'_{(2)}$ are empirical measure of samples in $\mathcal{D}_{te} = \{X^{(2)}, Y^{(2)}\}$ by randomly permuting the $n_{X,(2)} + n_{Y,(2)}$ many binary class labels. Since permuting test labels does not affect $\hat{g}_{(1)}(x)$, the test-only bootstrap does not require retraining of the neural network nor revisiting the training samples. Alternatively, one can permute the binary class labels in both $\mathcal{D}_{tr}$ and $\mathcal{D}_{te}$, and this will require to retain the neural network to obtain the new witness function $\hat{g}_{(1)}$ given the new class labels of $\mathcal{D}_{tr}$. We call such a bootstrap the *full-bootstrap*. The full-bootstrap can be applied to the symmetric MMD statistic without test set splitting as well, namely the setting of Theorem 3.1, to obtain an estimate of optimal $t_{\text{thres}}$ in practice.

We give two theoretical results on the testing power guarantee of the asymmetric NTK-MMD statistic (16): For *test-only bootstrap*, Theorem 3.2 proves testing power by restricting to good events over the randomness of $\mathcal{D}_{tr}$; For *full bootstrap*, the guarantee is provided in Theorem 3.3, which is the counterpart of Theorem 3.1. All proofs are in Appendix B.

We assume the balance-ness of the two samples as well as the training and testing splitting, that is, $n_{X,(1)}/n_X \to \rho_{X,(1)}$, $n_{Y,(1)}/n_Y \to \rho_{Y,(1)}$ $n_X/n \to \rho_X$, and the three constants are all in $(0, 1)$. With $n = n_X + n_Y$, we assume that for a constant $0 < c_a < 1$,

$$c_a n \le n_{X,(i)}, n_{Y,(i)} \le n, \quad i = 1, 2. \tag{19}$$

We denote by $\mathbb{P}_{(1)}$ the randomness over $\mathcal{D}_{tr}$, and $\mathbb{P}_{(2)}$ that over $\mathcal{D}_{te}$.

**Theorem 3.2** (Test power of $\hat{T}_a$, test-only bootstrap)**.** *Suppose that* (11), (19) *and the conditions (i) and (ii) in Theorem 3.1 hold, and* $0 < \gamma < 1$ *is a small number. Define* $\lambda_{(2),1} := \sqrt{4 \log(4/\alpha_{\text{level}})}$, $\lambda_{(1)} := \sqrt{4 \log(8/\gamma)}$, *and set the threshold as* $t_{\text{thres}} = 4(\sqrt{1.1}\lambda_{(2),1} + \lambda_{(1)})\sqrt{\nu/(c_a n)}$. *If $n$ is large enough such that* $n > (\frac{\lambda_{(1)}}{0.1\nu})^2/(8c_a)$, *and for some* $\lambda_{(2),2} > 0$,

$$n > \frac{1}{c_a} \max\left\{ \frac{1}{9\nu} \max\{\lambda_{(1)}, \lambda_{(2),1}, \lambda_{(2),2}\}^2, \frac{16\nu}{\delta_K^2} \left( 2\lambda_{(1)} + \sqrt{1.1}(\lambda_{(2),1} + \lambda_{(2),2}) \right)^2 \right\}, \tag{20}$$

*then, under both $H_0$ and $H_1$ there is a good event over the randomness of $\mathcal{D}_{tr}$ which happens w.p.$\ge 1 - \gamma$, under which, conditioning on $\mathcal{D}_{tr}$, $\mathbb{P}_{(2)}[\hat{T}_a > t_{\text{thres}}] \le \alpha_{\text{level}}$ under $H_0$, and $\mathbb{P}_{(2)}[\hat{T} \le t_{\text{thres}}] \le 4e^{-\lambda_{(2),2}^2/4}$ under $H_1$.*

*Remark* 3.2 (Sampling complexity). Compared to the full-bootstrap result Theorem 3.3, the additional requirement on $n$ is that $c_a n$ needs to be greater than $(\lambda_{(1)}/\nu)^2$ up to absolute constant, and thus when $\nu/\delta_K^2 \gg \nu^{-2}$, the $(\nu/\delta_K^2)$-term still dominates the needed lower bound of $n$, same as in Theorems 3.1 and 3.3. (Here we treat $\lambda_{(1)}$, $\lambda_{(2),1}$ and $\lambda_{(2),2}$ as $O(1)$ constants. Because the constant $\gamma$ controls the good event probability over the randomness of $\mathcal{D}_{tr}$, thus if $\gamma$ can be chosen to be of the same order as $\alpha_{\text{level}}$, then $\lambda_{(1)}$ has the same order as $\lambda_{(2),1}$.) The result shows that with test split and test-only bootstrap (avoiding retraining), the test power has the same order of needed sampling complexity, $n \gtrsim \nu/\delta_K^2$, as full bootstrap, with high probability and for large enough $n$.

**Theorem 3.3** (Test power of $\hat{T}_a$, full bootstrap). *Suppose that* (11), (19) *and the conditions (i) and (ii) in Theorem 3.1 hold. Define* $\lambda_1 := \sqrt{8 \log(4/\alpha_{\text{level}})}$, *and let the threshold for the test be* $t_{\text{thres}} = 4\lambda_1 \sqrt{\frac{\nu}{c_a n}}$. *Then, if for some* $\lambda_2 > 0$, $n$ *is large enough such that*

$$n > \frac{1}{c_a} \max\left\{ \frac{1}{9\nu} \max\{\lambda_1, \lambda_2\}^2, \ \frac{\nu}{\delta_K^2} \left(4(\lambda_1 + \lambda_2)\right)^2 \right\}, \tag{21}$$

*then under* $H_0$, $\mathbb{P}[\hat{T} > t_{\text{thres}}] \leq \alpha_{\text{level}}$; *and under* $H_1$, $\mathbb{P}[\hat{T} \leq t_{\text{thres}}] \leq 4e^{-\lambda_2^2/8}$.

**Pilot data setting.** This section considers the setting where we may have many samples for one distribution, e.g., the $p$. For instance, in change-point detection, where we are interested in detecting a shift in the underlying data distribution, there can be a large pool of pilot data before the change happens, collected historically and representing the normal status. We may have fewer data samples for the distribution $q$. For such a case, we can use data from the reference pool represent distribution $p$ to train the model and calibrate the threshold, e.g., using bootstrap. Since such "training" is done offline, we can afford the higher computational cost associated with training the model multiple times. In short, our strategy is to pre-compute the detector (retrain multiple times) and then use boostrap to obtain the threshold $t_{\text{thres}}$ for detector: (i) compute the symmetric MMD $\hat{T}$ as in (9) on $\{\hat{p}, \hat{q}\}$, where $\hat{q}$ is the new coming test samples (e.g. in change-point detection), and $\hat{p}$ is from the pool; (ii) pre-compute the symmetric MMD $\hat{T}_{\text{null}}$ on $\{\hat{p}_2, \hat{p}_2'\}$ from the pool of samples, with retraining, and obtain the "true" threshold for $\hat{T}$. Retraining of the network is expensive, but this is pre-computation and not counted in the online computation.

## 4 Numerical experiments

The section presents several experiments to examine the proposed method and validate the theory. [1]

### 4.1 Gaussian mean and covariance shifts

**Set-up**. Consider Gaussian mean shift and covariance shift in $\mathbb{R}^{100}$, where $n_X = n_Y = 200$; $p$ is the distribution of $\mathcal{N}(0, I_d)$, $d = 100$: (i) Mean-shift: $q$ is the distribution of $\mathcal{N}(\mu, I_d)$, where $\|\mu\|_2 = \delta$ which varies from 0 to 0.8 and (ii) Covariance-shift: $q$ is the distribution of $\mathcal{N}(0, I_d + \rho E)$, where $E$ is an $d$-by-$d$ all-ones matrix, and $\rho$ changes from 0 to 0.16. We split training and test sets into halves, and compute the asymmetric network approximated NTK-MMD statistic $\hat{T}_{a,\text{net}}$ (17), and estimate the test threshold by the quantile of (18); $H_0$ is rejected if $\hat{T}_{a,\text{net}} > t_{\text{thres}}$. We use a 2-layer network (1 hidden layer) with soft-plus activation. The online training is of 1 epoch (1 pass over the training set) with batch-size = 1. The bootstrap estimate of test threshold uses $n_{\text{boot}} = 400$ permutations. The testing power is approximated by $n_{\text{run}} = 500$ Monte Carlo replicas, and we compare with the benchmarks by (i) Hotelling's T-test, and (ii) Gaussian kernel MMD test (median distance bandwidth) [23]. The median distance bandwidth is a reasonable choice for detecting high dimensional Gaussian mean shift [38]. Both Gaussian kernel MMD and Hotelling's Test have access to all the samples $\mathcal{D}_{tr} \cup \mathcal{D}_{te}$. More experimental details are in Appendix C.1.

**Results**. The results are shown in the left two plots in Figure 1. The NTK-MMD test gives comparable but slightly worse power than the other two benchmarks on the mean shift. On the covariance shift test, the network MMD test gives equally good power as the Gaussian MMD. For the Gaussian covariance shift case, we also compute the testing power when only part of the training samples are used in the online training, and the results are shown in the right two plots in Figure 1. Testing power increases as the neural network scans more training samples, and when the covariance shifts are larger the transition takes place with smaller training sample size.

---

[1]Code available at `https://github.com/xycheng/NTK-MMD/`.

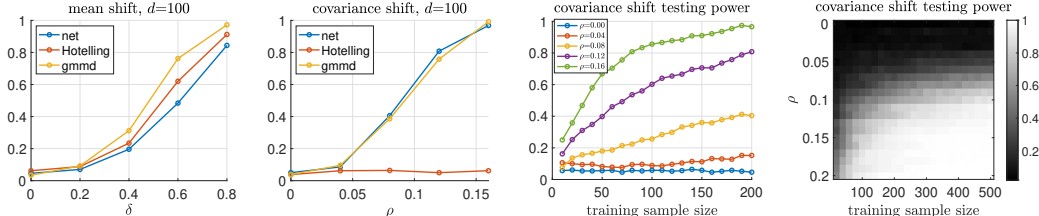

Figure 1: (Left two plots) Estimated testing power on Gaussian mean shift (change size is $\delta$) and Gaussian covariance shift (change size is $\rho$) in $\mathbb{R}^{100}$, where datasets $X$ and $Y$ have 200 samples respectively, and the training and testing splitting is half-half, i.e. $n_{tr} = n_{te} = 200$. Test power is estimated from $n_{run} = 500$. (Right two plots) Estimated testing power as a function of the number of samples processed in the 1-pass of the training set (batch size =1) and over varying values of $\rho$, also plotted as a color field.

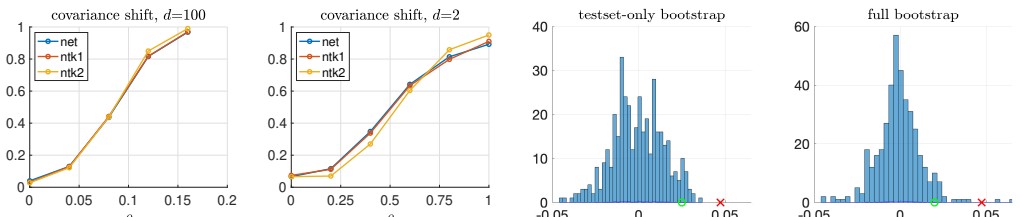

Figure 2: (Left two plots) Estimated testing power from $n_{\mathrm{run}} = 500$ of the covariance shift test in Figure 1 in $\mathbb{R}^{100}$ and $\mathbb{R}^2$. $n_X = n_Y = 200$, using three statistics: $\hat{T}_{\mathrm{net}}$ (net), $\hat{T}_{\mathrm{NTK}}$ with test set only bootstrap (ntk1) and with full bootstrap (ntk2) the training and testing splitting is half-half. (Right two plots) Test statistics $\hat{T}_a$ (red cross), the empirical distribution of $\hat{T}_{a,\mathrm{null}}$ using the test-only bootstrap and the full bootstrap (blue bars), and the estimated threshold (green circle). Computed from NTK kernel at $t = 0$ and $n_{\mathrm{boot}} = 400$.

In addition, we show in Appendix C.2 that NTK-MMD gives similear performance with varying network architectures, activation functions (like relu), and SGD configurations, and possibly better testing power with a larger network depth and width (Tables A.1 and A.2). We also compare with linear-time kernel MMD in Appendix C.7. As shown in Table A.3, NTK-MMD outperforms linear-time gMMD as in [23, Section 6], and underperforms the full gMMD which however requires $O(n^2)$ computation and storage.

### 4.2 Comparison of $\hat{T}_{\mathrm{net}}$ and $\hat{T}_{\mathrm{NTK}}$

**Set-up**. Since we use a 2-layer fully-connected network, the finite-width NTK at $t = 0$ (using initialized neural network parameters) can be analytically computed, which gives an $n_{te}$-by-$n_{tr}$ asymmetric kernel matrix $K$. The expression of $K$ and more details are provided in Appendix C.3. This allows computing the exact NTK-MMD (16), as well as the (i) full bootstrap and (i) the test-only bootstrap of the MMD statistic under $H_0$ by (i) permuting both rows and columns simultaneously and (ii) only permuting rows of the matrix $K$.

**Results.** To verify the $O(t)$ discrepancy as in Proposition 2.1, we first compute the numerical values of $\hat{T}_{\mathrm{NTK}}$ and $\hat{T}_{\mathrm{net}}(t)$ for different values of $t$ (which corresponds to different learning rate $\alpha$ as explained in Remark 2.2 and Appendix A.2) and the relative approximation error defined as $\mathrm{err} = |\hat{T}_{\mathrm{net}}(t) - \hat{T}_{\mathrm{NTK}}|/|\hat{T}_{\mathrm{NTK}}|$. The results are shown in Figure A.1. The fitted scaling of the error for softplus activation is about 0.96, which agrees with the theoretical $O(t)$ error. Switching to relu, the order is not close to 1 (instead 0.62) but $\hat{T}_{\mathrm{net}}(t)$ still gives a good approximation of $\hat{T}_{\mathrm{NTK}}$ as the relative error achieves about $10^{-3}$. The comparison of the testing power of network approximate NTK statistic and the exact NTK statistic tests are shown in Figure 2. In the high dimensional Gaussian covariance shift test ($d = 100$), the powers of the three tests are similar. When reducing dimension to $d = 2$, the full-bootstrap NTK tests show slightly different testing power than the other two. The network approximate NTK and NTK with test-only bootstrap always show almost the same testing power, consistent with the theory in Subsection 2.3. In the experiment on $\mathbb{R}^2$ data, the estimated threshold by full-bootstrap is smaller than by test-only bootstrap (right two plots), which explains the possibly better-testing power.

### 4.3 Comparison to neural network classification two-sample tests

**Set-up**. We experimentally compare NTK-MMD and state-of-the-art classification two-sample test (C2ST) baselines, which are neural network based tests. Following [36], we compare with C2ST-S,

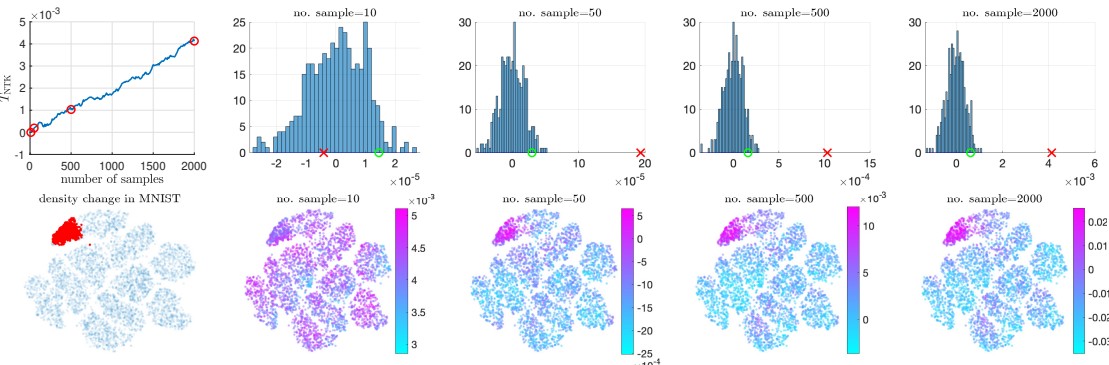

Figure 3: NTK-MMD statistic to detect distribution abundance change in MNIST digit image data; $n_{tr} = 2000$ and $nte$ is about 4000. (Top) Most left: Change of the statistic over the number of samples in the online training (batch size =1) of a 2-layer convolutional network. From 2nd-5th columns: The MMD statistic $\hat{T}_a$ compared with the empirical distribution under $H_0$ via test-only bootstrap, at four times along the online training (red circles on the left plot). (Bottom) Most left: The change of distribution of MNIST dataset embedded in 2D by tSNE [46]. From 2nd-5th columns: The witness function $\hat{g}$ plotted as a color field over the samples, at the four times corresponding to the upper panel plots.

| $n_{tr}$ | 2000 | 4000 | 6000 | 8000 |
|---|---|---|---|---|
| ME* | $\sim 10.0$ | $\sim 30.0$ | $\sim 58.0$ | $\sim 75.0$ |
| SCF* | $\sim 5.0$ | $\sim 6.0$ | $\sim 10.0$ | $\sim 15.0$ |
| C2ST-S (Adam) | 9.9 (61.6) | 14.0 (95.8) | 39.1 (100.0) | 61.2 (100.0) |
| C2ST-L (Adam) | 14.1 (87.8) | 38.4 (100.0) | 76.4 (100.0) | 92.9 (100.0) |
| C2ST-S (SGD) | 6.0 (13.9) | 10.6 (50.0) | 10.8 (94.4) | 14.8 (99.6) |
| C2ST-L (SGD) | 6.7 (22.2) | 12.8 (81.6) | 22.1 (100.0) | 34.6 (100.0) |
| NTK-MMD | 7.1 | 9.6 | 13.7 | 17.9 |

Table 1: Test power on Gaussian mixture data Example 1, dimension $d = 10$. (*recovered from Figure 3 in [36], $\sim$ means about.) For C2ST's, the number outside brackets is for epoch = 1, and in brackets for epoch = 10.

the classification accuracy test [37], and C2ST-L, the classification logit test [10]. Experimental details are given in Appendix C.4. The data distributions are:

 - Example 1: Gaussian mixture, fixed dimension $d = 10$ and increasing $n_{tr}$, which is the same setting as Figure 3 (left 2 plots) in [36]. Numbers in Table 1 show testing power (in %).

 - Example 2: Modified Gaussian mixture (from Example 1), the covariance shift is $I + 0.1E$ in both mixtures, where $E$ is all-one matrix with zeros on the diagonal. Dimension $d = 10$, and number of training samples $n_{tr}$ increases. The test power is shown in Table 2.

**Results**. On Example 1, NTK-MMD performs similar to SCF test in most cases, better than C2ST-S (SGD 1-epoch), and is worse than the other baselines. On Example 2, NTK-MMD outperforms C2ST baselines in several cases, e.g., constantly better than C2ST-S (SGD and Adam, 1-epoch) and comparable to C2ST-L (SGD 1-epoch). Note that C2ST baselines can be sensitive to training hyperparameters, such as the choice of optimization algorithm (SGD or Adam) and number of epochs. As far as the authors are aware of, there is no theoretical training guarantee of C2ST tests. In contrast, NTK-MMD has theoretical training guarantees due to the provable approximation to a kernel MMD. The weakness of NTK-MMD, though, is that the NTK kernel may not be discriminative to distinguish certain distribution departures, like in Example 1. The expressiveness power of NTK-MMD may be theoretically analyzed, for example, in the infinite-width limit using the analytical formula, as the infinite-width NTK has been shown to be universal for data on hyperspheres [28]. Overall, the results suggest that the performances of the three neural network tests depend on the data distributions, which is anticipated for any hypothesis test. Further theoretical investigations are postponed here.

| $n_{tr}$ | 500 | 1000 | 1500 | 2000 |
|---|---|---|---|---|
| C2ST-S (Adam) | 21.8 (28.1) | 62.2 (53.8) | 79.4 (74.0) | 94.6 (85.2) |
| C2ST-L (Adam) | 48.5 (49.4) | 92.8 (82.6) | 99.5 (96.3) | 100.0 (98.8) |
| C2ST-S (SGD) | 7.4 (28.3) | 22.7 (79.7) | 35.3 (92.4) | 54.9 (96.8) |
| C2ST-L (SGD) | 18.3 (52.2) | 56.8 (97.6) | 81.4 (99.9) | 97.3 (100.0) |
| NTK-MMD | 34.3 | 68.9 | 88.8 | 95.9 |

Table 2: Test power on Gaussian mixture data Example 2, dimension $d = 10$.

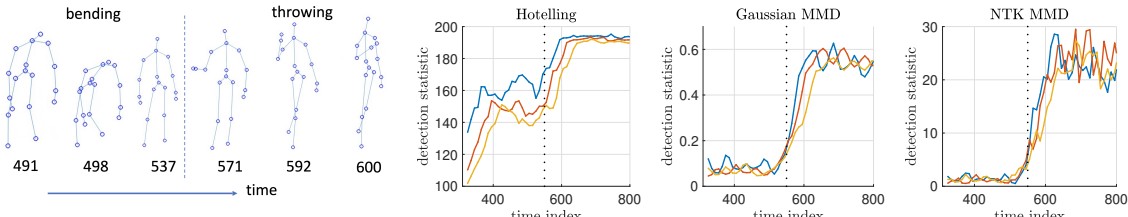

Figure 4: (Left) Example data sequence in the human activity dataset: before and after the change-point. (Right) Detection statistics computed by Hotelling's T statistic, Gaussian MMD, and NTK-MMD, on human action trajectory dataset, using window size 100 (blue), 150 (red), and 200 (yellow), respectively. The change point is at time index 550 (dotted black).

## 4.4 MNIST distribution abundance change

**Dataset**. We take the original MNIST dataset, which contains $28 \times 28$ gray-scale images, and construct two densities $p$ and $q$ by subsampling from the 70000 images in 10 classes, following [12]: $p$ is uniformly subsampled from the MNIST dataset, $p = p_{\text{data}}$, and $q$ has a change of abundance $q = 0.85 p_{\text{data}} + 0.15 p_{\text{cohort}}$, where $p_{\text{cohort}}$ is the distribution of a subset of the class of digit "1" having about 1900 samples. The $p_{\text{cohort}}$ is illustrated in the left bottom plot in Figure 3. The two samples $X$ and $Y$ have $n_X = 3000$, $n_Y = 2981$, and we randomly split $X$ and $Y$ make the training set $\mathcal{D}_{tr} = \{X_{(1)}, Y_{(1)}\}$, $n_{X,(1)} = n_{Y,(1)} = 1000$, and the rest is the test set $\mathcal{D}_{te}$.

**Results**. Using a 2-layer convolutional nerual network, we compute the network MMD statistic $\hat{T}_{a,\text{net}}$ (17) and the test-only bootstrap (18). The online training uses batch size =1 and one epoch, and more experimental details are in Appendix C.5. The results are shown in Figure 3. The NTK-MMD statistic already shows testing power after being trained on 50 samples, and in the later stage of training, the NTK witness function $\hat{g}_{(1)}$ identifies the region of the abundance change.

## 4.5 Online human activity change-point detection

**Set-up**. We present an illustrative example using NTK-MMD test statistic to perform online change-point detection: detecting human activity transition. We consider a real-world dataset, the Microsoft Research Cambridge-12 (MSRC-12) Kinect gesture dataset [18]. The data sequence records a human subject repetitively bending the body/picking up and throwing a ball before/after the change happens. After preprocessing, the sequential dataset contains 1192 frames (samples) and 54 attributes (data samples are in $\mathbb{R}^{54}$), with a change of action from "bending" to "throwing" at time index 550. More description of the dataset and experimental details is provided in Appendix C.6. Example samples before and after the change point are shown in the left of Figure 4. The algorithm is based on a sliding window which moves forward with time, and we compute the detection statistic every ten frames; such a procedure can be viewed as the Shewhart Chart in the literature [47]; scanning MMD statistic has been used in [32]. The window size is chosen to be 100, 150, and 200, respectively. We take a block of data (same size as the window) before the time index 300 (to use as the pilot samples) and compare with the distribution of data from the sliding window to compute the detection statistic. If there is a change-point, the detection statistic will show a large value.

**Results**. The other two detection statistics are computed by (i) Gaussian MMD (with bandwidth chosen to be median distance) and (ii) Hotelling's T statistics. The results are shown in Figure 4, where both the Gaussian MMD and the NTK-MMD statistics can detect the change: the detection statistic value remains low before the change and remains high after the change point, and both are better than the Hotelling statistic.

## 5 Discussion

The current work can naturally be extended in several aspects. First, the analysis of NTK approximation error may be extended, e.g., to other activation functions, and under the infinite-width limit. Second, considering other training objectives may allow us to compare NTK-MMD to other neural network classification tests. At the same time, the limitation of lazy-regime training has been studied in [21, 22, 39], which indicates that NTK theory cannot fully characterize the modeling ability of deep networks. It has also been shown that the expressiveness of the NTK kernel may be restricted to certain limited type of kernels [8, 20, 9]. This motivates extensions of NTK for studying deep network training [26, 41]. Finally, the application may extend to various hypothesis testing tasks as well as deep generative models. We thus view the current work as a first step towards understanding the role and potential of trained neural networks in testing problems and applications.

## Acknowledgement

The authors thank Alexander Cloninger and Galen Reeves for helpful discussion on the initial version of the paper, and the anonymous reviewers for helpful feedback. The work is supported by NSF DMS-2134037. XC is partially supported by NSF, NIH and the Alfred P. Sloan Foundation. YX is partially supported by NSF (CAREER Award 1650913, DMS-1938106, and DMS-1830210).

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
