# A    Proofs and additional analysis in Section 2

## A.1    Proofs in Section 2.3

*Remark* A.1 (Biased and unbiased MMD estimators). The exact NTK-MMD statistic (9) is a biased estimator [23]. The unbiased estimator is by excluding the diagonal terms of kernel matrix in the summation and normalizing by "$1/n(n-1)$" instead of "$1/n^2$". We consider biased estimator for simplicity, and also because the testing power analysis gives similar results, c.f. the comment beneath Theorem 3.1. In addition, the "asymmetric MMD statistic" (16) (with training-test splitting and used in many practical situations) is an unbiased estimator.

*Proof of Lemma 2.1.* By (3),

$$\frac{\partial \hat{L}}{\partial \theta} = - \int_{\mathcal{X}} \nabla_\theta f(x; \theta)(\hat{p} - \hat{q})(x)dx. \tag{22}$$

By the GD training dynamic,

$$\frac{\partial}{\partial t} u(x, t) = \langle \nabla_\theta f(x, \theta(t)), \dot{\theta}(t) \rangle \tag{23}$$

$$= \left\langle \nabla_\theta f(x, \theta(t)), -\frac{\partial \hat{L}}{\partial \theta} \right\rangle \tag{24}$$

$$= \int_{\mathcal{X}} \langle \nabla_\theta f(x; \theta(t)), \nabla_\theta f(x'; \theta(t)) \rangle (\hat{p} - \hat{q})(x')dx'. \tag{25}$$

This proves the lemma by definition of $\hat{K}_t(x, x')$ in (6). $\qquad\square$

*Proof of Lemma 2.2.* To prove Part (1): The initial weights $\theta(0) \in \Theta$, and by Taylor expansion,

$$\theta(t) = \theta(0) + t\dot{\theta}(t'),$$

from $0 < t' < t < t_{f,r}$. By definition,

$$\dot{\theta}(t') = -\frac{\partial \hat{L}}{\partial \theta}\bigg|_{\theta = \theta(t')} = \int_{\mathcal{X}} \nabla_\theta f(x; \theta(t'))(\hat{p} - \hat{q})(x)dx, \tag{26}$$

and thus by that $\|\nabla_\theta f\|_{\mathcal{X}, \Theta} \leq L_f$.

$$\|\dot{\theta}(t')\| \leq 2L_f,$$

This give that $\|\theta(t) - \theta(0)\| \leq t2L_f$, which proves Part (1).

To prove Part (2): Note that for any $x, x'$, and $t < t_{f,r}$,

$$\frac{\partial}{\partial t} \hat{K}_t(x, x') = R_t(x, x') + R_t(x', x), \quad R_t(x, x') := \langle \mathrm{D}_\theta^2 f(x, \theta(t))(\dot{\theta}(t)), \nabla_\theta f(x', \theta(t)) \rangle.$$

By Taylor expansion, for some $0 < t' < t$,

$$\hat{K}_t(x, x') = K_0(x, x') + t(R_{t'}(x, x') + R_{t'}(x', x))$$

where by part (1), $\theta(t') \in B_r$. Again by (26), this gives that

$$\|\dot{\theta}(t')\| \leq 2\|\nabla_\theta f\|_{\mathcal{X}, B_r},$$

where note that the domain of $\theta$ is $B_r \subset \Theta$, and thus the constant $\|\nabla_\theta f\|_{\mathcal{X}, B_r}$ can potentially be smaller than $L_f$. Then,

$$\|\mathrm{D}_\theta^2 f(x, \theta(t'))(\dot{\theta}(t'))\| \leq 2\|\mathrm{D}_\theta^2 f\|_{\mathcal{X}, B_r} \|\nabla_\theta f\|_{\mathcal{X}, B_r}.$$

As a result,

$$|R_{t'}(x, x')| \leq \|\mathrm{D}_\theta^2 f(x, \theta(t'))(\dot{\theta}(t'))\| \|\nabla_\theta f(x', \theta(t'))\| \leq 2\|\mathrm{D}_\theta^2 f\|_{\mathcal{X}, B_r} \|\nabla_\theta f\|_{\mathcal{X}, B_r}^2.$$

The same bound holds for $|R_{t'}(x, x')|$, and the above bounds are uniformly for all $x, x'$. Putting together, this proves Part (2). $\qquad\square$

*Proof of Proposition 2.1.* By definition,

$$\hat{T}_{net}(t) - \hat{T}_{\text{NTK}} = \int_{\mathcal{X}} \int_{\mathcal{X}} \frac{1}{t} \int_0^t \left( \hat{K}_s(x, x') - K_0(x, x') \right) ds (\hat{p} - \hat{q})(x') dx' (\hat{p} - \hat{q})(x) dx$$

$$= \int_{\mathcal{X}} \int_{\mathcal{X}} E(x, x') \hat{p}(x') \hat{p}(x) dx' dx - \int_{\mathcal{X}} \int_{\mathcal{X}} E(x, x') \hat{p}(x') \hat{q}(x) dx' dx$$

$$- \int_{\mathcal{X}} \int_{\mathcal{X}} E(x, x') \hat{q}(x') \hat{p}(x) dx' dx + \int_{\mathcal{X}} \int_{\mathcal{X}} E(x, x') \hat{q}(x') \hat{q}(x) dx' dx \qquad (27)$$

where we define

$$E(x, x') := \frac{1}{t} \int_0^t \left( \hat{K}_s(x, x') - K_0(x, x') \right) ds.$$

By Lemma 2.2, for $t < t_{f,r}$ and for any $x, x' \in \mathcal{X}$

$$|E(x, x')| \le \frac{1}{t} \int_0^t \left| \hat{K}_s(x, x') - K_0(x, x') \right| ds \le \frac{1}{t} \int_0^t C_{f,r} s \, ds = \frac{t}{2} C_{f,r}.$$

Thus the four terms in (27) in absolute value are all upper bounded by $C_{f,r} t/2$, and thus $|\hat{T}_{net}(t) - \hat{T}_{\text{NTK}}|$ is upper bounded by the sum of the absolute values of the four terms which is less than or equal to $2 C_{f,r} t$. $\qquad \square$

## A.2 Extension to SGD training

Consider the online setting of training the network by minimizing the loss $\hat{L}(\theta)$ in (3) on $n = n_X + n_Y$ samples. We write the training set $\mathcal{D}_{tr} = \{(z_i, l_i)\}_{i=1}^n$, where $z_i$ is from $X$ or $Y$, and $l_i = 1$ or 2 is the class label. Let $b_i = 1/n_X$ if $l_i = 1$, and $1/n_Y$ if $l_i = 2$. The loss can be written as

$$\hat{L}(\theta) = \sum_{i=1}^n f(z_i; \theta) b_i, \quad b_i = \begin{cases} -1/n_X, & l_i = 1, \\ 1/n_Y, & l_i = 2. \end{cases}$$

For simplicity, assume that $n_X = n_Y = n/2$. We define $l_i(\theta) = b_i f(z_i; \theta)$, which is the loss from the $i$-th sample.

Suppose we train the network with batch size =1 and 1 epoch. The learning rate is $\alpha$, that is, for $k$-th iteration in the SGD, $k = 1, \cdots, n$,

$$\theta_k = \theta_{k-1} - \alpha \nabla_\theta l_k(\theta_{k-1}),$$

from some $\theta_0 \in \Theta$. Note that $\nabla_\theta l_k(\theta) = b_k \nabla_\theta f(z_k; \theta)$, and thus

$$\|\theta_k - \theta_{k-1}\| = \alpha \|\nabla_\theta l_k(\theta_{k-1})\| = \alpha |b_k| \|\nabla_\theta f(z_k; \theta_{k-1})\| \le 2 L_f \frac{\alpha}{n}. \qquad (28)$$

This implies that

$$\|\theta_k - \theta_0\| \le 2 L_f \frac{k}{n} \alpha, \qquad (29)$$

and in particular, $\|\theta_n - \theta_0\| \le 2 L_f \alpha$. Thus, $\theta_k$ for all $k$ up to $n$ stays in a $r$-Euclidean ball of $\theta_0$ if $2 L_f \alpha < r$.

We write the network function at $k$-th step as $u_k$, $u_k(x) = f(x; \theta_k)$.

$$u_k(x) - u_{k-1}(x) = f(x; \theta_k) - f(x; \theta_{k-1})$$

$$= \nabla_\theta f(x; \theta_{k-1})^T (\theta_k - \theta_{k-1}) + O(\|\theta_k - \theta_{k-1}\|^2)$$

$$= -\alpha b_k \nabla_\theta f(x; \theta_{k-1})^T \nabla_\theta f(z_k; \theta_{k-1}) + O\left( (\frac{\alpha}{n})^2 \right), \qquad (30)$$

where we treat $L_f$ as $O(1)$ constant, and the same with other constants which depend on the infinity norm of derivatives of $f$.

We analyze how $\nabla_\theta f(x; \theta_{k-1})^T \nabla_\theta f(z_k; \theta_{k-1})$ differs from $\nabla_\theta f(x; \theta_0)^T \nabla_\theta f(z_k; \theta_0)$. For any $x \in \mathcal{X}$,

$$\nabla_\theta f(x; \theta_{k-1}) = \nabla_\theta f(x; \theta_0) + O(\|\theta_{k-1} - \theta_0\|),$$

and by (29),

$$\nabla_\theta f(x; \theta_{k-1}) = \nabla_\theta f(x; \theta_0) + O\left(\frac{k-1}{n}\alpha\right).$$

Thus,

$$\nabla_\theta f(x; \theta_{k-1})^T \nabla_\theta f(z_k; \theta_{k-1}) = \left\langle \nabla_\theta f(x; \theta_0) + O\left(\frac{k-1}{n}\alpha\right), \nabla_\theta f(x; \theta_0) + O\left(\frac{k-1}{n}\alpha\right) \right\rangle$$

$$= \nabla_\theta f(x; \theta_0)^T \nabla_\theta f(z_k; \theta_0) + O\left(\frac{k-1}{n}\alpha\right).$$

Back to (30), we have

$$u_k(x) - u_{k-1}(x) = -\alpha b_k \left(\nabla_\theta f(x; \theta_0)^T \nabla_\theta f(z_k; \theta_0) + O\left(\frac{k-1}{n}\alpha\right)\right) + O\left((\frac{\alpha}{n})^2\right)$$

$$= -\alpha b_k \nabla_\theta f(x; \theta_0)^T \nabla_\theta f(z_k; \theta_0) + O\left(\frac{k-1}{n^2}\alpha^2\right) + O\left((\frac{\alpha}{n})^2\right).$$

This give that

$$u_n(x) - u_0(x) = -\alpha \sum_{k=1}^n b_k \nabla_\theta f(x; \theta_0)^T \nabla_\theta f(z_k; \theta_0) + \sum_{k=1}^n O\left(k\frac{\alpha^2}{n^2}\right)$$

$$= \alpha \int_{\mathcal{X}} K_0(x, x')(\hat{p} - \hat{q})(x')dx' + O(\alpha^2),$$

where recall that $K_0(x, x') = \nabla_\theta f(x; \theta_0)^T \nabla_\theta f(x'; \theta_0)$ is the NTK at time zero. This proves that

$$\hat{g}(x) := \frac{1}{\alpha}(u_n(x) - u_0(x)) = \int_{\mathcal{X}} K_0(x, x')(\hat{p} - \hat{q})(x')dx' + O(\alpha).$$

Comparing to the continuous time training dynamic, we see that $\alpha$ corresponds to training time $t$, and with batch size 1 the SGD training the NTK approximation has the same $O(\alpha)$ error as with the continuous time GD training.

## B  Proofs and additional theoretical results in Section 3

### B.1  Proofs in Subsection 3.1

The proof of Theorem 3.1 uses the U-statistic concentration analysis, which was used in Theorem 3.5 in [12]. The analysis in [12] is for the local RBF kernel, and we need to extend to the general PSD kernel here.

The concentration argument is by Proposition B.1. Note that the concentration can be derived using the boundedness (11) alone, while the Bernstein-type control here is sharper when the squared integrals upper bound $\nu$ is much smaller than 1.

**Proposition B.1** (Concentration of $\hat{T}_{\mathrm{NTK}}$). *Assuming* (11), (14) *and the conditions (i) and (ii) in Theorem 3.1,*

*(1) Under $H_0$, when $0 < \lambda < 3\sqrt{c\nu n}$, w.p. $\geq 1 - 3e^{-\lambda^2/8}$, $\hat{T} \leq \frac{4}{cn} + 4\lambda\sqrt{\frac{\nu}{cn}}$.*

*(2) Under $H_1$, when $0 < \lambda < 3\sqrt{c\nu n}$, w.p. $\geq 1 - 3e^{-\lambda^2/8}$, $\hat{T} \geq \delta_K - 4\lambda\sqrt{\frac{\nu}{cn}}$.*

The proof of Theorem 3.1 is a direct application of the proposition.

*Proof of Theorem 3.1.* Note that condition (15) ensures that

$$\max\{\lambda_1, \lambda_2\} < 3\sqrt{\nu cn}, \quad \frac{4}{cn} < 0.5\delta_K, \quad 4(\lambda_1 + \lambda_2)\sqrt{\frac{\nu}{cn}} < 0.5\delta_K, \tag{31}$$

and the bounds in Proposition B.1 parts (1) and (2) hold with $\lambda_1$ and $\lambda_2$ respectively.

To verify that $\mathbb{P}[\hat{T} > t_{\text{thres}}] \leq \alpha_{\text{level}}$ under $H_0$: Observe that $3e^{-\lambda_1^2/8} = \alpha_{\text{level}}$ by the definition of $\lambda_1$, and then the claim follows by Proposition B.1 Part (1) since $\lambda_1 < 3\sqrt{\nu cn}$.

To bound $\mathbb{P}[\hat{T} \leq t_{\text{thres}}]$ under $H_1$: Since $\lambda_2 < 3\sqrt{\nu cn}$, by Proposition B.1 Part (1), the claim holds if

$$t_{\text{thres}} = \frac{4}{cn} + 4\lambda_1\sqrt{\frac{\nu}{cn}} < \delta_K - 4\lambda_2\sqrt{\frac{\nu}{cn}}, \tag{32}$$

which is guaranteed by (31). □

*Remark* B.1 (Asymptotic choice of $t_{\text{thres}}$). The optimal $t_{\text{thres}}$ in Theorem 3.1 as the $(1 - \alpha_{\text{level}})$-quantile of the distribution of $\hat{T}$ under $H_0$ can be obtained potentially analytically according to the limiting distribution of the MMD statistic: The asymptotic distribution of (squared) empirical MMD statistic has been derived using the spectral decomposition of the (centered) kernel function $\tilde{k}(x, x') := K(x, x') - \mathbb{E}_{y \sim p}K(x, y) - \mathbb{E}_{y \sim p}K(y, x') + \mathbb{E}_{y, y' \sim p}K(y, y')$ in [23, 11], among others, following techniques in Chapter 6 in [42]. Specifically, by Theorem 3.3 in [11], as $n = n_X + n_Y \to \infty$ and $n_X/n \to \rho_X \in (0, 1)$, $n\hat{T}$ under $H_0$, $q = p$, converges in distribution to the weighted $\chi^2$ distribution $\sum_{k=1}^{\infty} \tilde{\lambda}_k \xi_k^2$, where $\xi_k \sim \mathcal{N}(0, 1/\rho_X + 1/(1 - \rho_X))$ i.i.d, and $\tilde{\lambda}_k \geq 0$ are the eigenvalues of the integral operator with kernel $\tilde{k}(x, x')$ in $L^2(\mathcal{X}, p(x)dx)$. This provides the asymptotic value of the quantile of $\hat{T}$ under $H_0$, when the eigenvalues are computable, which can be useful, e.g., for low-dimensional data.

*Proof of Proposition B.1.* The proof follows the approach in Proposition. 3.4 in [12]. By definition,

$$\hat{T} := \frac{1}{n_X^2}\sum_{i,j=1}^{n_X} K(x_i, x_j) + \frac{1}{n_Y^2}\sum_{i,j=1}^{n_Y} K(y_i, y_j) - \frac{2}{n_X n_Y}\sum_{i=1}^{n_X}\sum_{j=1}^{n_Y} K(x_i, y_j), \tag{33}$$

and equivalently,

$$\hat{T} = \hat{T}_{X,X} + \hat{T}_{Y,Y} - 2\hat{T}_{X,Y}, \tag{34}$$

$$\hat{T}_{X,X} = \frac{1}{n_X^2}\sum_{i,j=1}^{n_X} K(x_i, x_j), \quad \hat{T}_{Y,Y} = \frac{1}{n_Y^2}\sum_{i,j=1}^{n_Y} K(y_i, y_j), \quad \hat{T}_{X,Y} = \frac{1}{n_X n_Y}\sum_{i=1}^{n_X}\sum_{j=1}^{n_Y} K(x_i, y_j). \tag{35}$$

The terms $\hat{T}_{X,X}$ and $\hat{T}_{Y,Y}$ contain diagonal entries of the kernel matrix which have different marginal distributions from the off-diagonal entries. Define

$$D_X := \frac{1}{n_X}\sum_{i=1}^{n_X} K(x_i, x_i), \quad V_{X,X} := \frac{1}{n_X(n_X - 1)}\sum_{i \neq j, i,j=1}^{n_X} K(x_i, x_j),$$

then

$$\hat{T}_{X,X} = \frac{1}{n_X}D_X + (1 - \frac{1}{n_X})V_{X,X} = V_{X,X} + \frac{1}{n_X}(D_X - V_{X,X}). \tag{36}$$

Observe that

$$|V_{X,X}| \leq \frac{1}{n_X(n_X - 1)}\sum_{i \neq j, i,j=1}^{n_X} |K(x_i, x_j)|$$

$$\leq \frac{1}{n_X(n_X - 1)}\sum_{i \neq j, i,j=1}^{n_X} \sqrt{K(x_i, x_i)}\sqrt{K(x_j, x_j)} \quad (K(x, x') \text{ is PSD})$$

$$\leq \frac{1}{n_X(n_X - 1)}\sum_{i \neq j, i,j=1}^{n_X} \frac{1}{2}(K(x_i, x_i) + K(x_j, x_j))$$

$$= \frac{1}{n_X}\sum_{i=1}^{n_X} K(x_i, x_i) = D_X,$$

and, in addition, by (11),
$$0 \le D_X \le 1.$$
Thus (36) gives that
$$V_{X,X} \le \hat{T}_{X,X} \le V_{X,X} + \frac{2}{n_X} D_X \le V_{X,X} + \frac{2}{n_X}. \tag{37}$$
The random variable $V_{X,X}$ is a U-statistic, where for $i \ne j$,
$$\mathbb{E}K(x_i, x_j) = \mathbb{E}_{x \sim p, y \sim p} K(x, y),$$
and by condition (ii),
$$\mathrm{Var}(K(x_i, x_j)) \le \mathbb{E}_{x \sim p, y \sim p} K(x, y)^2 = \nu_{p,p} \le \nu.$$
As for the boundedness of the r.v. $K(x_i, x_j)$, by (11),
$$|K(x_i, x_j)| \le 1 = L.$$
By the de-coupling of U-statistic in Proposition. 3.4 in [12], we obtain the Bernstein-type control of the tail probability, that is
$$\mathbb{P}\left[V_{X,X} - \mathbb{E}_{x \sim p, y \sim p} K(x, y) > t\right] \le \exp\{-\frac{\frac{n_X - 1}{2} t^2}{2\nu + \frac{2}{3} tL}\}, \quad \forall t > 0.$$

Let $t = \lambda \sqrt{\frac{\nu}{n_X - 1}}$, to obtain the sub-Gaussian tail we need $tL < 3\nu$, that is, $t < 3\nu$ by that $L = 1$. This gives that when $0 < \lambda < 3\sqrt{\nu(n_X - 1)}$,
$$\mathbb{P}\left[V_{X,X} - \mathbb{E}_{x \sim p, y \sim p} K(x, y) > \lambda \sqrt{\frac{\nu}{n_X - 1}}\right] \le \exp\{-\frac{(n_X - 1)t^2}{8\nu}\} = e^{-\lambda^2/8}.$$
The same holds for $\mathbb{P}\left[V_{X,X} - \mathbb{E}_{x \sim p, y \sim p} K(x, y) < -\lambda \sqrt{\frac{\nu}{n_X - 1}}\right]$. Meanwhile, by (14),
$$cn \le n_X - 1.$$
Together with (37), this gives that when $0 < \lambda < 3\sqrt{\nu cn}$,
$$\hat{T}_{X,X} \le \mathbb{E}_{x \sim p, y \sim p} K(x, y) + \lambda \sqrt{\frac{\nu}{cn}} + \frac{2}{cn}, \quad \text{w.p.} \ge 1 - e^{-\lambda^2/8},$$
$$\hat{T}_{X,X} \ge \mathbb{E}_{x \sim p, y \sim p} K(x, y) - \lambda \sqrt{\frac{\nu}{cn}}, \quad \text{w.p.} \ge 1 - e^{-\lambda^2/8}. \tag{38}$$

The similar bound can be proved for $\hat{T}_{Y,Y}$, by defining $D_Y$ and $V_{Y,Y}$ similarly, and using that $\nu_{q,q} \le \nu$ and $cn \le n_Y - 1$.

To analyze the concentration of $\hat{T}_{X,Y}$, which consists of the summation over the $n_X$-by-$n_Y$ array, the de-coupling argument gives that for $M := \min\{n_X, n_Y\}$, and any $0 < \lambda < 3\sqrt{\nu M}$,
$$\mathbb{P}\left[\hat{T}_{X,Y} > \mathbb{E}_{x \sim p, y \sim q} K(x, y) + \lambda \sqrt{\frac{\nu}{M}}\right] \le e^{-\lambda^2/8},$$
and same for $\mathbb{P}\left[\hat{T}_{X,Y} < \mathbb{E}_{x \sim p, y \sim q} K(x, y) - \lambda \sqrt{\frac{\nu}{M}}\right]$. By that $cn \le M$, when $0 < \lambda < 3\sqrt{\nu cn}$,
$$\hat{T}_{X,Y} \le \mathbb{E}_{x \sim p, y \sim q} K(x, y) + \lambda \sqrt{\frac{\nu}{cn}}, \quad \text{w.p.} \ge 1 - e^{-\lambda^2/8},$$
$$\hat{T}_{X,Y} \ge \mathbb{E}_{x \sim p, y \sim q} K(x, y) - \lambda \sqrt{\frac{\nu}{cn}}, \quad \text{w.p.} \ge 1 - e^{-\lambda^2/8}. \tag{39}$$

Finally, to prove Part (1) of the proposition, use the upper bound in (38), the corresponding upper bound for $\hat{T}_{YY}$, and the lower bound in (39). This gives that, when $0 < \lambda < 3\sqrt{\nu cn}$, under the intersection of the three good events, which happens w.p. $\ge 1 - 3e^{-\lambda^2/8}$, we have that
$$\hat{T}_{X,X} + \hat{T}_{Y,Y} - 2\hat{T}_{X,Y} \le (\mathbb{E}_{x \sim p, y \sim p} + \mathbb{E}_{x \sim q, y \sim q} - 2\mathbb{E}_{x \sim p, y \sim q}) K(x, y) + 4\lambda \sqrt{\frac{\nu}{cn}} + \frac{4}{cn},$$
where the first term vanishes since $p = q$ under $H_0$. To prove part (2), use the lower bounds in (38), in the counterpart of (38) for $\hat{T}_{YY}$, and the upper bound in (39). $\qquad\square$

## B.2 Proof of Theorem 3.2

In the proof of Theorem 3.2 and 3.3 which involves training and testing splitting, we use subscript $_{(1)}$ to denote the randomness over $\mathcal{D}_{tr}$, and subscript $_{(2)}$ that over $\mathcal{D}_{te}$, possibly conditioned on $\mathcal{D}_{tr}$. We use the notations $\mathbb{P}_{(i)}$, $\mathbb{E}_{(i)}$ and $\text{Var}_{(i)}$, for $i = 1, 2$. We say $E$ is a good event in $\mathbb{P}_{(1)}$ which happens w.p. $\geq 1 - \delta$ in $\mathbb{P}_{(1)}$ if $\mathbb{P}_{(1)}[E^c] \leq \delta$, where $0 < \delta < 1$ is a small number.

Theorem 3.2 is based on Lemma B.1 which establishes the concentration of the conditional expectation $\mathbb{E}[\hat{T}_a | \mathcal{D}_{tr}]$, and Proposition B.2 on the concentration of $\hat{T}_a$ under good events of $\mathcal{D}_{tr}$.

*Proof of Theorem 3.2.* We first consider under $H_0$, where $\delta_K = 0$. Let $\gamma = 8\delta$, and applying Lemma B.1 with $\lambda_{(1)}$ such that

$$e^{-\lambda_{(1)}^2/4} = \delta,$$

which gives the same value of $\lambda_{(1)}$ as in the statement of the theorem. We have that there is a good event $E_1$ in $\mathbb{P}_{(1)}$, which happens w.p. $\geq 1 - 4\delta$, such that under $E_1$,

$$\hat{C} \leq \delta_K + 4\lambda_{(1)}\sqrt{\frac{\nu}{c_a n}} = 4\lambda_{(1)}\sqrt{\frac{\nu}{c_a n}}, \tag{40}$$

and this requires

$$\lambda_{(1)} < 3\sqrt{\nu c_a n}. \tag{41}$$

Applying Proposition B.2 (1), there is another good event $E_2$ in $\mathbb{P}_{(1)}$, which happens w.p. $\geq 1 - 4\delta$, such that under $E_2$,

$$\mathbb{P}_{(2)}\left[\hat{T}_a > \hat{C} + 4\lambda_{(2),1}\sqrt{\frac{1.1\nu}{c_a n}}\right] \leq 4e^{-\lambda_{(2),1}^2/4}, \tag{42}$$

as long as

$$\lambda_{(2),1} < 3\sqrt{1.1\nu c_a n}, \quad \sqrt{\log(1/\delta)/(2c_a n)} = \sqrt{\lambda_{(1)}^2/(8c_a n)} \leq 0.1\nu. \tag{43}$$

We thus set

$$4e^{-\lambda_{(2),1}^2/4} = \alpha_{\text{level}}, \quad t_{\text{thres}} = 4\lambda_{(1)}\sqrt{\frac{\nu}{c_a n}} + 4\lambda_{(2),1}\sqrt{\frac{1.1\nu}{c_a n}},$$

which gives the same values of $\lambda_{(2),1}$ and $t_{\text{thres}}$ as in the statement of the theorem. Then, under the intersection event $E_1 \cap E_2$ which happens w.p. $\geq 1 - 8\delta = 1 - \gamma$ in $\mathbb{P}_{(1)}$, combining (40) and (42) gives that

$$\mathbb{P}[\hat{T}_a > t_{\text{thres}}] \leq \alpha_{\text{level}}.$$

Next, under $H_1$, similarly, there are good events $E_1'$ and $E_2'$, the intersection of which happens w.p. $\geq 1 - \gamma$ in $\mathbb{P}_{(1)}$, and under $E_1' \cap E_2'$,

$$\hat{C} \geq \delta_K - 4\lambda_{(1)}\sqrt{\frac{\nu}{c_a n}},$$

and

$$\mathbb{P}_{(2)}\left[\hat{T}_a < \hat{C} - 4\lambda_{(2),2}\sqrt{\frac{1.1\nu}{c_a n}}\right] \leq 4e^{-\lambda_{(2),2}^2/4},$$

and this requires

$$\lambda_{2,(2)} < 3\sqrt{1.1\nu c_a n}. \tag{44}$$

This means that the Type-II error bound under $H_1$ in the theorem holds as long as

$$\delta_K - 4\lambda_{(1)}\sqrt{\frac{\nu}{c_a n}} - 4\lambda_{2,(2)}\sqrt{\frac{1.1\nu}{c_a n}} > t_{\text{thres}}. \tag{45}$$

Collecting the needed requirements (41) (43) (44) (45), and they are satisfied by (20) and the assumption of the theorem. $\square$

In both Lemma B.1 and Proposition B.2, suppose that (11), (19) and the conditions (i) and (ii) in Theorem 3.1 hold. We define the witness function of exact NTK-MMD as

$$\hat{g}_{\text{NTK}}(x) := \int_{\mathcal{X}} K(x, x')(\hat{p}_{(1)} - \hat{q}_{(1)})(x')dx'. \tag{46}$$

**Lemma B.1.** *Denote the conditional expectation $\mathbb{E}[\hat{T}_a | \mathcal{D}_{tr}]$ as*

$$\hat{C} := \int_{\mathcal{X}} \hat{g}_{\mathrm{NTK}}(x)(p - q)(x)dx, \tag{47}$$

*then for any $0 < \lambda_{(1)} < 3\sqrt{\nu c_a n}$,*

$$\mathbb{P}_{(1)}\left[\hat{C} - \delta_K > 4\lambda_{(1)}\sqrt{\frac{\nu}{c_a n}}\right] \leq 4e^{-\lambda_{(1)}^2/4},$$

*and same with $\mathbb{P}_{(1)}[\hat{C} - \delta_K < -4\lambda_{(1)}\sqrt{\frac{\nu}{c_a n}}]$.*

**Proposition B.2.** *Suppose $0 < \delta < 1$ and $\sqrt{\log(1/\delta)/(2c_a n)} \leq 0.1\nu$, then under both $H_0$ and $H_1$, there is a good event which happens w.p. $\geq 1 - 4\delta$ over the randomness of $\mathcal{D}_{tr}$, under which, conditioning on $\mathcal{D}_{tr}$,*

*(1) Under $H_0$, $\mathbb{P}_{(2)}[\hat{T}_a > \hat{C} + 4\lambda\sqrt{\frac{1.1\nu}{c_a n}}] \leq 4e^{-\lambda^2/4}$ if $0 < \lambda < 3\sqrt{1.1\nu c_a n}$;*

*(2) Under $H_1$, $\mathbb{P}_{(2)}[\hat{T}_a < \hat{C} - 4\lambda\sqrt{\frac{1.1\nu}{c_a n}}] \leq 4e^{-\lambda^2/4}$ if $0 < \lambda < 3\sqrt{1.1\nu c_a n}$.*

*Proof of Proposition B.2.* In this proof we write $\hat{g}_{\mathrm{NTK}}$ defined in (46) as $\hat{g}$ for shorthand notation. We have that $\hat{g} = \hat{g}_X - \hat{g}_Y$, where

$$
\begin{aligned}
\hat{g}_X(x) &:= \int_{\mathcal{X}} K(x, x')\hat{p}_{(1)}(x')dx' = \frac{1}{n_{X,(1)}} \sum_{i=1}^{n_{X,(1)}} K(x, x_i^{(1)}), \\
\hat{g}_Y(x) &:= \int_{\mathcal{X}} K(x, x')\hat{q}_{(1)}(x')dx' = \frac{1}{n_{Y,(1)}} \sum_{i=1}^{n_{Y,(1)}} K(x, y_i^{(1)}),
\end{aligned}
\tag{48}
$$

and both $\hat{g}_X$ and $\hat{g}_Y$ are determined by $\mathcal{D}_{tr}$. By definition,

$$
\begin{aligned}
\hat{T}_a &= \int_{\mathcal{X}} (\hat{g}_X - \hat{g}_Y)(x)(\hat{p}_{(2)} - \hat{q}_{(2)})(x)dx \\
&= \frac{1}{n_{X,(2)}} \sum_i \hat{g}_X(x_i^{(2)}) - \frac{1}{n_{Y,(2)}} \sum_i \hat{g}_X(y_i^{(2)}) - \frac{1}{n_{X,(2)}} \sum_i \hat{g}_Y(x_i^{(2)}) + \frac{1}{n_{Y,(2)}} \sum_i \hat{g}_Y(x_i^{(2)}) \\
&:= S_{X,X} - S_{X,Y} - S_{Y,X} + S_{Y,Y}.
\end{aligned}
\tag{49}
$$

Conditioning on a realization of $\mathcal{D}_{tr}$, due to the independence of $\mathcal{D}_{te}$ from $\mathcal{D}_{tr}$, the four terms in (49) are independent sums of random variables over the randomness of $\mathcal{D}_{te}$. Again, we analyze the concentration of these four terms respectively, conditioned on $\mathcal{D}_{tr}$ and we will restrict to good events in $\mathbb{P}_{(1)}$.

We start from $S_{X,X}$. Again by (11), we have $|\hat{g}_X(x)| \leq 1$ for any $x \in \mathcal{X}$. Meanwhile, $\forall x \in \mathcal{X}$,

$$\hat{g}_X(x)^2 = \left(\frac{1}{n_{X,(1)}} \sum_i K(x, x_i^{(1)})\right)^2 \leq \frac{1}{n_{X,(1)}} \sum_i K(x, x_i^{(1)})^2,$$

and thus, conditioning on $\mathcal{D}_{tr}$,

$$\mathrm{Var}_{(2)}(\hat{g}_X(x_i^{(2)})) \leq \mathbb{E}_{x \sim p}\hat{g}_X(x)^2 \leq \frac{1}{n_{X,(1)}} \sum_i \mathbb{E}_{x \sim p}K(x, x_i^{(1)})^2 = \frac{1}{n_{X,(1)}} \sum_i \psi_p(x_i^{(1)}) =: \hat{\nu}_{p,X},$$
$$\tag{50}$$

where we define

$$\psi_p(x') := \int_{\mathcal{X}} K(x, x')^2 p(x)dx,$$

and $\hat{\nu}_{p,X}$ is a random variable determined by $\mathcal{D}_{tr}$. One can verify that by restricting to large probability event in $\mathbb{P}_{(1)}$, $\hat{\nu}_{p,X}$ concentrates at the mean value

$$\mathbb{E}_{(1)}\hat{\nu}_{p,X} = \int_{\mathcal{X}} \psi_p(x')p(x')dx' = \int_{\mathcal{X}} \int_{\mathcal{X}} K(x, x')^2 p(x)dx p(x')dx' = \nu_{p,p} \leq \nu. \tag{51}$$

Specifically, (11) implies that $0 \leq \psi_p(x') \leq 1$, and then by Hoeffding's inequality,

$$\mathbb{P}_{(1)}[\hat{\nu}_{p,X} - \mathbb{E}_{(1)}\hat{\nu}_{p,X} > t] \leq e^{-2n_{X,(1)}t^2} \leq e^{-2c_a nt^2}, \quad \forall t > 0.$$

Let $e^{-2c_a nt^2} = \delta$, where $\delta$ is as in the statement of the proposition, then w.p. $\geq 1 - \delta$ in $\mathbb{P}_{(1)}$,

$$\hat{\nu}_{p,X} \leq \mathbb{E}_{(1)}\hat{\nu}_{p,X} + t = \mathbb{E}_{(1)}\hat{\nu}_{p,X} + \sqrt{\frac{\log(1/\delta)}{2c_a n}} \leq \nu + 0.1\nu, \tag{52}$$

and the last inequality is by (51) and the condition of the proposition. We call this good event $E_{X,X}$ in $\mathbb{P}_{(1)}$, under which (52) holds.

Back to $S_{X,X}$, we have that under $E_{X,X}$ in in $\mathbb{P}_{(1)}$, and conditioning on the realization of $\mathcal{D}_{tr}$, $\hat{g}_X(x_i^{(2)})$ as r.v. in $\mathbb{P}_{(2)}$ are bounded as $|\hat{g}_X(x_i^{(2)})| \leq 1$; Meanwhile, by (50) and (52),

$$\text{Var}_{(2)}(\hat{g}_X(x_i^{(2)})) \leq \hat{\nu}_{p,X} \leq 1.1\nu.$$

Then the classical Bernstein gives that $\forall 0 < \lambda < 3\sqrt{1.1\nu n_{X,(2)}}$,

$$\mathbb{P}_{(2)}\left[S_{X,X} - \mathbb{E}_{(2)}S_{X,X} > \lambda\sqrt{\frac{1.1\nu}{n_{X,(2)}}}\right], \mathbb{P}_{(2)}\left[S_{X,X} - \mathbb{E}_{(2)}S_{X,X} < -\lambda\sqrt{\frac{1.1\nu}{n_{X,(2)}}}\right] \leq e^{-\lambda^2/4}.$$

By that $n_{X,(2)} \geq c_a n$, we have that $\forall 0 < \lambda < 3\sqrt{1.1\nu c_a n}$, under the good event $E_{X,X}$ which happens w.p. $\geq 1 - \delta$ in $\mathbb{P}_{(1)}$ and conditioning on $\mathcal{D}_{tr}$,

$$\mathbb{P}_{(2)}\left[S_{X,X} - \mathbb{E}_{(2)}S_{X,X} > \lambda\sqrt{\frac{1.1\nu}{c_a n}}\right], \mathbb{P}_{(2)}\left[S_{X,X} - \mathbb{E}_{(2)}S_{X,X} < -\lambda\sqrt{\frac{1.1\nu}{c_a n}}\right] \leq e^{-\lambda^2/4}. \tag{53}$$

Similarly, we can show that, there are good events $E_{X,Y}$, $E_{Y,X}$, and $E_{Y,Y}$ over randomness of $\mathcal{D}_{tr}$, where each happens in $\mathbb{P}_{(1)}$ w.p. $\geq 1 - \delta$, and under which the similar bound as (53) holds for $S_{X,Y}$, $S_{Y,X}$, and $S_{Y,Y}$ respectively as long as $0 < \lambda < 3\sqrt{1.1\nu c_a n}$. Thus, under the intersection of the four good events, which happens in $\mathbb{P}_{(1)}$ w.p. $\geq 1 - 4\delta$,

$$\mathbb{P}_{(2)}\left[\hat{T}_a - \mathbb{E}_{(2)}\hat{T}_a > 4\lambda\sqrt{\frac{1.1\nu}{c_a n}}\right], \mathbb{P}_{(2)}\left[\hat{T}_a - \mathbb{E}_{(2)}\hat{T}_a < -4\lambda\sqrt{\frac{1.1\nu}{c_a n}}\right] \leq 4e^{-\lambda^2/4}.$$

The above holds under both $H_0$ and $H_1$. Finally, by that $\mathbb{E}_{(2)}\hat{T}_a = \hat{C}$ as defined in (47), this proves parts (1) and (2) of the proposition. $\square$

*Proof of Lemma B.1.* Note that $\hat{C}$ is a random variable over the randomness of $\mathcal{D}_{tr}$ only. By definition,

$$\hat{C} = \int_{\mathcal{X}}\int_{\mathcal{X}} K(x,x')(\hat{p}_{(1)} - \hat{q}_{(1)})(x')dx'(p - q)(x)dx = \int_{\mathcal{X}}(\varphi_p - \varphi_q)(x')(\hat{p}_{(1)} - \hat{q}_{(1)})(x')dx',$$

where

$$\varphi_p(x') := \int_{\mathcal{X}} K(x,x')p(x)dx, \quad \varphi_q(x') := \int_{\mathcal{X}} K(x,x')q(x)dx.$$

Because only $n_{X,(1)}$ and $n_{Y,(1)}$ are involved here, in this proof we write $n_{X,(1)}$ as $n_X$ and $n_{Y,(1)}$ as $n_Y$ for notation convenience, and we also denote samples from $X_{(1)}$ and $Y_{(1)}$ by $x_i$ and $y_i$ respectively. By (19), we then have

$$n_X, n_Y \geq c_a n. \tag{54}$$

We then equivalently write $\hat{C}$ as

$$\hat{C} = \frac{1}{n_X}\sum_{i=1}^{n_X}\varphi_p(x_i) - \frac{1}{n_X}\sum_{i=1}^{n_X}\varphi_q(x_i) - \frac{1}{n_Y}\sum_{i=1}^{n_Y}\varphi_p(y_i) + \frac{1}{n_Y}\sum_{i=1}^{n_Y}\varphi_q(y_i)$$
$$:= C_{X,X} - C_{X,Y} - C_{Y,X} + C_{Y,Y}, \tag{55}$$

and we use concentration argument on the four terms respectively.

Due to (11),
$$|\varphi_p(x)| \leq 1, \quad |\varphi_q(x)| \leq 1, \quad \forall x \in \mathcal{X}.$$

Starting from $C_{X,X}$ which is an independent sum of i.i.d. rv's, where $|\varphi_p(x_i)| \leq 1 := L$; By that

$$\varphi_p(x)^2 = \left( \int_{\mathcal{X}} K(x,x')p(x')dx' \right)^2 \leq \left( \int_{\mathcal{X}} K(x,x')^2 p(x')dx' \right) \left( \int_{\mathcal{X}} p(x')dx' \right) = \int_{\mathcal{X}} K(x,x')^2 p(x')dx'$$

we have

$$\mathrm{Var}_{(1)}(\varphi_p(x_i)) \leq \mathbb{E}_{x \sim p} \varphi_p(x)^2 \leq \int_{\mathcal{X}} \int_{\mathcal{X}} K(x,x')^2 p(x')dx' p(x)dx = \nu_{p,p} \leq \nu,$$

where the last inequality is by condition (ii) in Theorem 3.1. The classical Bernstein then gives that $\forall 0 < \lambda < 3\sqrt{\nu n_X}$,

$$\mathbb{P}_{(1)} \left[ C_{X,X} - \mathbb{E}_{(1)} C_{X,X} > \lambda \sqrt{\frac{\nu}{n_X}} \right], \ \mathbb{P}_{(1)} \left[ C_{X,X} - \mathbb{E}_{(1)} C_{X,X} < -\lambda \sqrt{\frac{\nu}{n_X}} \right] \leq e^{-\lambda^2/4}.$$

The similar bounds can be derived for $C_{X,Y}$, and for $C_{Y,X}$ and $C_{Y,Y}$ where $n_X$ is replaced with $n_Y$. By (54), this gives that when $0 < \lambda < 3\sqrt{\nu c_a n} \leq 3\sqrt{\nu n_X}$ and $3\sqrt{\nu n_Y}$,

$$\mathbb{P}_{(1)} \left[ \hat{C} - \mathbb{E}_{(1)} \hat{C} > 4\lambda \sqrt{\frac{\nu}{c_a n}} \right], \ \mathbb{P}_{(1)} \left[ \hat{C} - \mathbb{E}_{(1)} \hat{C} < -4\lambda \sqrt{\frac{\nu}{c_a n}} \right] \leq 4e^{-\lambda^2/4}.$$

Observing that $\mathbb{E}_{(1)} \hat{C} = \delta_K$ which is defined in (12) finishes the proof. □

## B.3 Test power of $\hat{T}_a$ with full-bootstrap

We derive here the testing power of the statistic $\hat{T}_a$ computed on split training/testing sets in Subsection 3.2, with a theoretical choice of $t_{\mathrm{thres}}$, similar to as in Theorem 3.1. In practice, the full-bootstrap estimation of $t_{\mathrm{thres}}$ can obtain better power than the theoretical one.

*Proof of Theorem 3.3.* Similar to the proof of Theorem 3.1 by applying Proposition B.3. Due to that the upper bound of $\hat{T}_a$ under $H_0$ does not have the $\frac{4}{cn}$ term, c.f. Proposition B.3 Part (1) (because the asymmetric kernel MMD is computed from an off-diagonal block of the kernel matrix and the summation in $\hat{T}_a$ does not involve diagonal terms), the value of $t_{\mathrm{thres}}$ does not have the $\frac{4}{cn}$ term, and the condition (21) has one term less on the r.h.s. than (15). □

**Proposition B.3** (Concentration of $\hat{T}_a$). *Assuming* (11), (19) *and the conditions (i) and (ii) in Theorem 3.1,*

*(1) Under $H_0$, when $0 < \lambda < 3\sqrt{c_a \nu n}$, w.p. $\geq 1 - 4e^{-\lambda^2/8}$, $\hat{T}_a \leq 4\lambda \sqrt{\frac{\nu}{c_a n}}$.*

*(3) Under $H_1$, when $0 < \lambda < 3\sqrt{c_a \nu n}$, w.p. $\geq 1 - 4e^{-\lambda^2/8}$, $\hat{T}_a \geq \delta_K - 4\lambda \sqrt{\frac{\nu}{c_a n}}$.*

The proof makes use of the independence of the four datasets $X_{(1)}$, $X_{(2)}$, $Y_{(1)}$ and $Y_{(2)}$, and the concentration of the double summation over the four blocks of the asymmetric kernel matrix.

*Proof of Proposition B.3.* By definition,

$$\hat{T}_a = \frac{1}{n_{X,(2)} n_{X,(1)}} \sum_{i=1}^{n_{X,(2)}} \sum_{j=1}^{n_{X,(1)}} K(x_i^{(2)}, x_j^{(1)}) - \frac{1}{n_{X,(2)} n_{Y,(1)}} \sum_{i=1}^{n_{X,(2)}} \sum_{j=1}^{n_{Y,(1)}} K(x_i^{(2)}, y_j^{(1)})$$
$$- \frac{1}{n_{Y,(2)} n_{X,(1)}} \sum_{i=1}^{n_{X,(1)}} \sum_{j=1}^{n_{Y,(2)}} K(x_i^{(1)}, y_j^{(2)}) + \frac{1}{n_{Y,(2)} n_{Y,(1)}} \sum_{i=1}^{n_{Y,(2)}} \sum_{j=1}^{n_{Y,(1)}} K(y_i^{(2)}, y_j^{(1)}).$$
(56)

Then, equivalently,

$$\hat{T}_a = T_{X,X} - T_{Y,X} - T_{X,Y} + T_{Y,Y} \tag{57}$$

$$T_{X,X} := \frac{1}{n_{X,(2)}n_{X,(1)}} \sum_{i=1}^{n_{X,(2)}} \sum_{j=1}^{n_{X,(1)}} K(x_i^{(2)}, x_j^{(1)}), \quad T_{Y,X} := \frac{1}{n_{X,(2)}n_{Y,(1)}} \sum_{i=1}^{n_{X,(2)}} \sum_{j=1}^{n_{Y,(1)}} K(x_i^{(2)}, y_j^{(1)}) \tag{58}$$

$$T_{X,Y} := \frac{1}{n_{Y,(2)}n_{X,(1)}} \sum_{i=1}^{n_{X,(1)}} \sum_{j=1}^{n_{Y,(2)}} K(x_i^{(1)}, y_j^{(2)}), \quad T_{Y,Y} := \frac{1}{n_{Y,(2)}n_{Y,(1)}} \sum_{i=1}^{n_{Y,(2)}} \sum_{j=1}^{n_{Y,(1)}} K(y_i^{(2)}, y_j^{(1)}). \tag{59}$$

We analyze the concentration of the four terms respectively, all similarly to the analysis of the "$\hat{T}_{X,Y}$" term in the proof of Proposition B.1, Specifically, for $T_{X,X}$: Define $M := \min\{n_{X,(1)}, n_{X,(2)}\}$, and by (19),

$$M \geq c_a n.$$

By that $\nu_{pp} \leq \nu$ and that the kernel is bounded in absolute value by 1, we have that $\forall 0 < \lambda < 3\sqrt{\nu M}$,

$$\mathbb{P}\left[T_{X,X} - \mathbb{E}_{x\sim p,y\sim p}K(x,y) > \lambda\sqrt{\frac{\nu}{M}}\right], \mathbb{P}\left[T_{X,X} - \mathbb{E}_{x\sim p,y\sim p}K(x,y) < -\lambda\sqrt{\frac{\nu}{M}}\right] \leq e^{-\lambda^2/8},$$

and $M$ can be replaced to be $c_a n$ where the claim remains to hold. Similar bounds hold for $T_{Y,X}$, $T_{X,Y}$, $T_{Y,Y}$, since

$$\min\{n_{X,(1)}, n_{X,(2)}, n_{Y,(1)}, n_{Y,(2)}\} \geq cn.$$

Putting together, to prove (1) under $H_0$, use the concentration bounds for the 4 quantities and under the joint good events, plus that $\mathrm{MMD}_K^2(p,q) = 0$. Part (2) under $H_1$ is proved similarly. □

## C Experimental details and additional results

### C.1 Gaussian mean and covariance shifts

The neural network has 2 fully-connected (fc) layers, i.e. 1 hidden layer, and has the following architecture: the input data dimension $d = 100$, the hidden layer width $m = 512$,

fc $(d, m)$ - softplus - fc $(m, 1)$ - loss as in (3)

where $(\mathrm{f}_{in}, \mathrm{f}_{out})$ stand for dimensionality of input and output features respectively.

The network mapping $f(x; \theta)$ can be equivalently written as

$$f(x; \theta) = \sum_{k=1}^{m} a_k \sigma(w_k^T x + b_k), \quad \theta = \{(w_k, b_k, a_k)\}_{k=1}^{m}. \tag{60}$$

The neural network parameters are initialized such that $a_k \sim \mathcal{N}(0, 1/m)$, $w_k \sim \mathcal{N}(0, I_d)$, and $b_k = 0$. For simplicity, we leave the 2nd layer parameters $a_k$ fixed after initialization and only train the 1st layer parameters $w_k$ and $b_k$.

*Remark* C.1 (Effective learning rate). The network is trained for 1 epoch (1 pass of the training set) and batch-size 1, using basic SGD. In the notation of Remark 2.2, the theoretical learning rate $\alpha = 0.1$. Note that the definition of loss (3) contains normalization $1/n_X$ and $1/n_Y$, and here $n_{X,(1)} = n_{Y,(1)} = 100$. Comparing to training objective which is usually defined as the summation (with out normalizing by sample size), the effective learning rate here (lr) is $\alpha/100 = 10^{-3}$. Using smaller values of lr produces similar results, but note that reducing lr to be too small may cause numerical issue, due to that the deep learning programs use single precision floating point arithmetic.

The testing powers are approximately computed over $n_{\mathrm{run}}$ random replicas. For Figure 1, the most right plot is produced by $n_{\mathrm{run}} = 200$, and all other plots by $n_{\mathrm{run}} = 500$. In the most right plot, $n_{X,(1)} = n_{Y,(1)} = 250$, and the effective lr is $0.1/250 = 4 \times 10^{-4}$.

| Neural network configuration \ width $m$ | 256 | 512 | 1024 |
|---|---|---|---|
| 2-layer softplus | 82.0 | 81.6 | 82.0 |
| 2-layer relu | 79.8 | 84.4 | 82.8 |
| 3-layer relu | 85.8 | 88.4 | 91.0 |

Table A.1: NTK-MMD with relu activation, different width $m$, and more layers (to compare to Figure 1, which is computed with 2 fc-layers, softplus activation, width $m$=512). Numbers in the table are testing power (in %).

| SGD configuration | Test power of NTK-MMD |
|---|---|
| Batch-size = 1, epoch= 1 (10) | 84.2 (85.0) |
| Batch-size = 20, epoch= 1 (10) | 82.8 (81.6) |

Table A.2: NTK-MMD trained with different numbers of epochs and batch sizes. The example of gaussian covariance shift in Section 4.1, $\rho = 0.12$. Test power (in %) with epoch=1 outside brackets, with epoch=10 in brackets.

## C.2 Experiments of varying neural network hyperparameters

We conducte additional experiments to investigate the influence of neural network architecture and training hyperparameters.

● Different activation functions, network depths and widths

Table A.1 shows that increasing the network depth can improve testing power, and changing from softplus to relu obtains similar results. We also find in experiments that relu can obtain more robustness of testing power performance with respect to different weight initialization schemes. We observe that the performance with wider networks is generally better, though no longer sensitive beyond a certain $m$. Theoretically, the convergence to infinite-width limiting NTK may lead to further analysis of the discriminative power of the kernel to distinguish $p$ and $q$, see the comments in Subsection 4.3.

● General SGD with varying batch-size, epochs, and batch-size

Theoretically, the analysis covers general SGD (more than one epoch and different batch size): The proof in Appendix A.2 generalizes to such cases because the residual error of the Taylor expansion of the network mapping $f(x; \theta)$ still applies.

Empirically, we verify that the testing power of NTK-MMD is not sensitive to batch size nor a few more epochs, as illustrated in Table A.2. This agrees with the theory that $\hat{T}_{net}$ computed with different batch-size and small number of epochs all approximate the exact NTK-MMD at time zero. In other experiments in the paper, we focus on batch-size =1 to show that NTK-MMD allows extremely small batch-size. Note that the advantage of NTK-MMD is particularly pronounced in the one-pass training, i.e., we can only visit the data in one-pass, which commonly appears in the streaming data setting.

## C.3 Computation of the exact NTK-MMD

The neural network setting is the same as in Subsection 4.1, and here we derive the expression of the NTK kernel at $t = 0$, which was used to compute the "ntk1" and "ntk2" statistics.

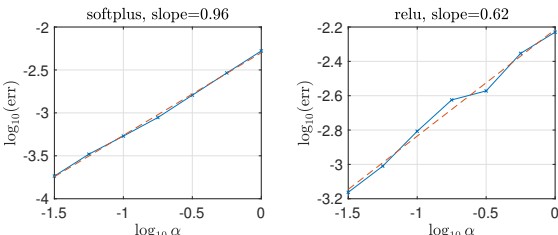

Figure A.1: Numerical computation of the (relative) approximation error err $= |\hat{T}_{net}(t) - \hat{T}_{NTK}|/|\hat{T}_{NTK}|$. The network has 2-fc layers with softplus (or relu) activation and width $m = 512$, and the simulated data distribution is Gaussian covariance shift in dimension 100.

For the network function as in (60),

$$K_0(x, x') = \left( \sum_{k=1}^{m} a_k^2 \sigma'(w_k^T x + b_k) \sigma'(w_k^T x' + b_k) \right) \left( 1 + x^T x' \right),$$ (61)

where $\sigma(z) = \log(1 + e^z)$ is the softplus function, and is differentiable on $\mathbb{R}$. Thus the kernel for any pair of samples $x$ and $x'$ is analytically computable once the network parameters are initialized. In our experiments, we compute $K_0(x, x')$ as in (61) with finite hidden-layer width $m$ and given realizations of the $t = 0$ network parameters.

## C.4 Comparison to neural network classification tests

The network is fc 3-layer with relu activation and width $m = 512$. Two C2ST baselines are trained with Adam and SGD respectively, and trained for 1 and 10 epochs. (By SGD, we mean vanilla SGD with constant step-size and no momentum.) NTK-MMD uses SGD, epoch $= 1$. We also experiment under $H_0$ to verify that the Type-I error achieves $\alpha_{\text{level}} = 0.05$.

Since [36] already compared C2ST's with optimization-based linear time kernel tests, namely ME and SCF tests [14, 29] and showed that C2ST's are generally better, we cite the results therein for comparison.

## C.5 MNIST distribution abundance change

The neural network has two convolutional (conv) layers:

conv 5x5x1x16 - relu - maxpooling 2x2

- conv 5x5x16x32 - relu - maxpooling 2x2

- fc ( ·,128) - relu - fc (128, 1) - loss

where the dimension of $f_{in}$ in the 1st fc layer is by flattening the input feature, which gives $f_{in} = 4^2 \cdot 32$ in this case.

In the online training of the network, we use batch size = 1, theoretical lr $\alpha = 0.01$, and SGD with momentum 0.9, Adding momentum to SGD is common in neural network practice, and we adopt it here as to examine the behavior of the model: theoretically, under the NTK assumption, we expect similar behavior with and without momentum in short-time training with SGD. As has been explained in Appendix C.1, by that $n_{X,(1)} = n_{Y,(1)} = 10^3$, the effective lr is $\alpha/10^3 = 10^{-5}$.

## C.6 Human activity change-point detection

The (MSRC-12) Kinect gesture dataset consists of sequences of human skeletal body part movements (represented as body part locations) collected from 30 people performing 12 gestures. There are 18 sensors in total, and each sensor records the coordinates in the three-dimensional Cartesian coordinate system at each time.

The net MMD statistic is computed using a 2-layer fc network having 512 hidden nodes and soft-plus activation. We use effective lr 0.0015 and SGD with momentum 0.9 in the 1-pass training with batch-size 1.

## C.7 Comparison to linear time MMD

The test power comparison of NTK-MMD, Gaussian kernel MMD and the linear-time version as in [23, Section 6], on the example of MNIST data in Section 4.4 is given in Table A.3. In both the full and linear-time Gaussian kernel MMD tests, median-distance kernel bandwidth is used, and the test has access to all the samples in training and testing sets (no splitting). On the examples in Section 4.1 (Figure 1) linear-time gaussian MMD baseline gives inferior power (all less than 10%, details omitted). This version of linear-time MMD only provides a global test statistic but not directly a witness function (to indicate where $p$ and $q$ differ), while NTK-MMD training obtains network witness function which approximates the kernel witness function of NTK.

As alternative linear-time MMD tests, the ME and SCF tests [14, 29] involve additional gradient-based optimization of model parameters and may not have optimization convergence guarantee for

| Test statistics $\setminus n_{tr}$ | 100 | 200 | 300 | 500 | 1000 | 2000 |
|---|---|---|---|---|---|---|
| *gmmd* | 62.0 | 93.2 | 99.6 | - | - | - |
| *gmmd-lin* | 7.0 | 10.8 | 12.6 | 16.0 | 24.4 | 36.4 |
| NTK-MMD | 35.4 | 67.6 | 86.2 | 98.2 | 100.0 | 100.0 |

Table A.3: Testing power (in %) of MNIST density departure example in Subsection 4.4. *gmmd* is Gaussian kernel MMD, and *gmmd-lin* the linear-time version. Results of *gmmd* for $n_{tr}$ greater than 300 are omitted due to slow computation.

general data distributions. NTK-MMD has comparable computational and memory complexity to classification neural network tests (the order is the same, but only one epoch is needed and batch size can be as small as one), and has learning guarantee via NTK approximation as shown in Section 2.3 and Section 3.