# OpenReview forum: "Neural Tangent Kernel Maximum Mean Discrepancy"
_NeurIPS.cc/2021/Conference — NeurIPS 2021 Poster_

### Official Review · Reviewer_Kznb · 2021-07-02

**Rating:** 3
**Confidence:** 4

**Summary:**

The paper addresses two-sample testing with a neural-network-based maximum mean discrepancy.
The paper essentially considers two types of test statistics: (a) MMD defined by a neural tangent kernel (NTK), and (b) mean differences based on neural network (NN) features.
For two-sample testing, the paper seems to consider the second kind only; the test is specifically a mean difference test, where data features are given by a NN trained on a held-out dataset.

**Main Review:**

# Overview
I appreciate the authors’ effort to develop a fast, high-power two-sample test with NN features.
The paper, however, needs improvement. In particular, the proposed tests lack empirical and theoretical justifications at multiple levels.

As described in Summary, the authors consider the mean-difference test (the test stat is defined in Eq. (17)) in their two-sample test experiments.
While the test is promising and could be improved with a more elaborate NN, the authors failed to show its utility in their experiments; We can see that the performance is comparable to the MMD test (Gretton et al., 2012) with the Gaussian kernel. Therefore, leaving aside its issues (discussed below) as a two-sample test, the paper is not practically compelling and not ready for publication.

For the other statistic, defined by the NTK of a trained network, the authors did not show its merits relative to a variety of existing approaches, making it difficult to value the paper's contribution. What makes the paper not appealing is the following:
* There is no comparison to other methods in the MNIST experiment
* Changepoint detection; apart from the lack of competitors, the behaviour of the test looks almost identical to the Gaussian kernel MMD.

# Detailed comments

## On the mean difference test (the test-only bootstrap)
In addition to the aforementioned weakness, the test lacks theoretical comparison.
Existing tests, such as the original MMD test and a deep kernel version  (Liu et al., 2020), are at least asymptotically guaranteed to distinguish any pair of distributions.
On the other hand, the proposed test lacks such property; the defined maximum mean discrepancy (or integral probability metric) is not guaranteed to separate two distributions (i.e., it is only zero if and only two distributions agree), and optimisation might get stuck in bad local optima.
Given that the empirical performance is not well demonstrated, it is desirable, while challenging, that the paper discusses when and how the test works better (e.g., what kind of alternatives the test is adept at detecting).
Also, if the paper's focus is on improving the time complexity of quadratic time MMD tests, it would be helpful to contrast the proposed approach with existing linear time tests such as ChwialkowskiJ et al., (2015); Jitkrittum et al., (2016). Currently, the paper lacks such obvious baselines.

Note that the power analysis (Theorem 3.2) addresses a different test; the analysed test uses a threshold defined by a concentration inequality, whereas the proposed test derives its threshold with permutation bootstrap. It would be helpful to state the relation between them.

## On the NTK test
The concentration analysis in Theorem 3.1 defines a test based on the NTK statistic. The analysis has the following issues:
* Constant normalisation of the kernel. The constant B is typically unknown for the NTK;  the subsequent concentration analysis implicitly depends on this constant, and the test may not be easy to implement in practice. Note that if we normalise the kernel $K$ as $K(x, y)/(\sqrt{K(x, x)}\sqrt{K(x, x)})$, we would need a different proof for the approximation (Eq. (10)) of the trained NTK with the NTK at initialisation.
* The NTK Approximation in Theorem 3.1 is not experimentally justified enough. The experiment considers a single null scenario, which makes the justification dubious.

Additionally, if the NTK test behaves like that of the NTK at initialisation without features learned,  the test’s benefit is unclear (how can it achieve higher power?)


## On the full-bootstrap test
This test was not compared to existing methods. Also, the proposed bootstrap method is not theoretically justified with proof as in the foregoing test-only bootstrap test. Its behaviour under the null is also not thoroughly investigated in experiments.

## Changepoint detection
It is unclear that the changepoint detection experiment highlights the proposed statistic. At the end of the day, we must determine if a change occurs, which requires a threshold. Merely tracking a statistic's value is not useful for this purpose.


===== post rebuttal =====

I thank the authors for their rebuttal. I misunderstood the proposed statistics at the time of writing my review, and the rebuttal clarified this. My evaluation on this work remains the same as the following questions are not satisfactorily answered:

1. As two-sample tests, are the proposed tests valid and useful?
2. Is the proposed statistic Eq. (5) useful for downstream applications?

On the first question, I doubt the validity of the test based on $\hat{T}_{net}$. The approximation by the NTK-MMD (Proposition 2) involves an unknown constant. If we determine the threshold by the NTK-MMD, we need to take into account the deviation caused by the approximation, which requires the constant. Even when the constant in Proposition 2 is available, it depends on the network architecture and initialisation and could be large, which could result in a conservative test. As the authors did not actually use the test in their experiments (and did not report the type-I errors of the test in the added experiments), the validity of the test is dubious.

Second, of the two bootstrap tests, I will focus on the test-only bootstrap as it is the test used in their experiments. The type-I guarantee for this test is okay unlike the first test discussed above. The test is not practically compelling:

* The reported results in Additional Table 3 do not show significant improvements by the test (see the linear time baseline ME).
* Additional Table 4 does show the superiority over the linear time MMD, but somehow it does not show other linear time baselines (including the ME test), which makes the result arbitrary.


 Is the proposed statistic Eq. (5) useful for downstream applications?
My understanding is that the authors base their argument on

1.  fast computation (linear-time)
2. online training (can be used for stream data)
3. learning guarantee (we can obtain globally optimal network params).

The first two points are definitely true; however, they apply to any neural network IPM like Eq. (3). It is unclear why the witness function of the form in Eq. (4) is desirable. For the third point, my impression is that this kind of argument is only applicable to shallow networks; even if we can obtain an optimal network within such a class, it is unclear how useful the witness function is (justification is not provided in the paper except for the approximation by the NTK-MMD). Apart from the optimisation consideration, overall, the paper lacks comparison in downstream tasks and fail to showcase the proposed method.

**Time Spent Reviewing:**

24 hours

---

### Official Review · Reviewer_e8cj · 2021-07-14

**Rating:** 4
**Confidence:** 4

**Summary:**

The paper presents a theoretical method to analyze the neural network-based two-sample test methods. In particular, it does so by identifying a correspondence between NTK and MMD

**Limitations And Societal Impact:**

Sufficiently addressed.

**Main Review:**

I think the idea is novel and the theory / empirical validation is solid, but I am concerned with the following two problems that I deem crucial.

(a) The NTK-MMD correspondence only applies to the loss function given in Eq.3, which is both artificial (not used in practice) and ill-defined (no global convergence). A study based on such kind of a loss function cannot convince me unless it is shown that this loss function can be useful in practice.

(b) The paper is also not sufficiently motivated in my opinion. In particular, I do not know how important neural network-based methods are for the problem of two-sample testing. For example, is it a very popular approach? Does it empirically achieve better performance than the traditional two-sample test methods? The paper does not say this, and I remain unconvinced about the importance of the neural network-based two-sample testing approach

In summary, I find the motivation and the setting of this paper contrived and of uncertain machine learning importance. I might consider raising the score if the authors convince me that Eq.3 is not as artificial as it seems by either showing that (1) it is useful (achieve SOTA) or that (2) standard two-sample testing loss functions can be approximated by this one (so that eq3 has some universal character to some extent) or (3) both. Problem (b) is easier to fix I guess, but the authors need to cite some relevant works to convince me (and other readers) that neural network-based approaches are indeed an important approach

**Time Spent Reviewing:**

1 hour

---

### Official Review · Reviewer_aee6 · 2021-07-15

**Rating:** 4
**Confidence:** 4

**Summary:**

This paper tried to apply the Neural Tangent Kernel (NTK) as the kernel function of MMD for a two-sample test task. Setting the MMD (without square; using hypothesis space of NN as the function space of MMD) as the training objective of the NN, the paper utilized the NTK which is trained after 1 epoch as the kernel function used in MMD. Theoretically, this paper provided the approximation error analysis for the NTK in the initialization state and the NTK after a short training and gave detailed bounds of NTK-MMD's testing power by deriving the sample complexity for a given level of Type-I error. Finally, the paper conducted a lot of experiments (on artificial datasets and real-world datasets) to verify the effectiveness of NTK-MMD.

**Limitations And Societal Impact:**

NAN

**Main Review:**

While the content is quite dense and solid, the paper is not that strongly motivated and lack of insights in both the theoretical and experimental parts. Some concerns are as follows.

1. Although the paper provided theoretical properties of NTK-MMD in Section.3, the derived bounds about testing power seem to have no specific relation with NTK. Besides, the paper did not compare the bounds of NTK with bounds of other common kernel functions and thus gave little insight. For example, is this testing power bound of NTK better than RBF/Laplacian kernel, or what's the contrast between them? If the bound holds for general PSD kernels, then why introduce NTK?

2. In the experiments, the paper only uses common finite neural networks, which is rather narrow compared to the needed infinite width that NTK/lazy training theory applies. In this narrow case, the parameters should change a lot after training, is it right?

Minor issue:
Page 4. Remark 2.2. (SGD and online training).

**Time Spent Reviewing:**

8 hours

---

### Official Review · Reviewer_tRT5 · 2021-07-17

**Rating:** 7
**Confidence:** 4

**Summary:**

The paper introduces a new neural network Maximum Mean Discrepancy statistic to achieve both computational and memory efficiency. The authors further characterize the statistical properties of the statistic including Type-I and Type-II errors. They also perform both simulation and real data analysis to address the efficiency of the proposed testing procedure.

**Limitations And Societal Impact:**

The authors adequately addressed the limitations and potential negative societal impact.

**Main Review:**

The main contribution of this paper lies in developing a novel neural network MMD statistic. It introduces the advantage of deep neural networks' representation and optimization to kernel MMD testing procedure, which solves the computational and memory difficulties of using MMD statistic. The observation that training of a network in a short time nearly means computing the witness function of a kernel MMD at time zero given some specific choice of objective function is insightful.

My minor concerns come from theoretical perspective. The analysis here seems to follow Cheng and Xie (2021). The authors may discuss more on the theoretical novelty. In addition, the paper only discusses the balanced sample setting. Some discussion on unbalanced data is appreciated, even from practical side.

**Time Spent Reviewing:**

6

---

### Decision · Program_Chairs · 2021-09-28

**Decision:**

Accept (Poster)

**Comment:**

In this paper, the authors proposed a theoretical method to analyze the neural network-based two-sample test methods. Though the paper is dense and solid, the motivation is not very clear. Some reviewers pointed out that theoretical insight was insufficient and numerical experiments were not convincing to show the effectiveness of the proposed method. Following the careful discussion, the reviewers agree that the paper is not yet ready for publication.

**Consistency Experiment:**

NeurIPS has a long history of experimentation. In 2014, NeurIPS ran an experiment in which 10% of submissions were reviewed by two independent committees to quantify the randomness in the review process. This year, we repeated a variant of this experiment to see how the quality of the review process has changed over time.  This paper was part of the experiment and was therefore assigned to two committees (consisting of reviewers, an Area Chair, and a Senior Area Chair) that reached independent decisions.  If both committees made the same recommendation, this recommendation was followed. If a single committee recommended acceptance, the paper was accepted (with the exception of a few cases in which the other committee identified what we considered a fatal flaw, e.g., an error in a key result).

This copy’s committee reached the following decision: **Reject**

The other committee assigned to the paper recommended **Accept (Poster)**.  You can find the other set of reviews, along with any follow up discussion with the authors here:
https://openreview.net/forum?id=i2pFtDzmPL6